# FORWARD-FORWARD LEARNING WITH DYNAMIC ARCHITECTURE ADAPTATION FOR CLASSIFICATION

## ABSTRACT

The Forward-Forward (FF) algorithm has emerged as a promising alternative to the traditional deep learning paradigm based on the backpropagation algorithm. However, both the original FF algorithm and several FF-based extensions rely on the quality of generated negative samples for training, which can limit their effectiveness. In this paper, we design an FF-based algorithm for the classification task. Specifically, we propose the concept of support neuron (SN) sets by partitioning the neurons in each layer into several sets, each explicitly corresponding to a class. The SN set with the strongest response (goodness) determines the predicted class of the input, thereby eliminating the need for negative samples. Furthermore, inspired by the functioning of the brain, we introduce neuron growth and degeneration strategies: (1) when neurons fail to achieve satisfactory performance, new neurons can grow to assist; and (2) neurons that remain inactive across all classes may degenerate. Extensive experiments demonstrate that our method achieves state-of-the-art performance on MNIST and CIFAR datasets compared to other FF-based approaches that also eliminate the use of negative samples. In addition, the effectiveness of the proposed neuron growth and degeneration mechanisms is empirically evaluated.

## 1 INTRODUCTION

Deep learning (Goodfellow et al., 2016) is a widely adopted technique that has achieved remarkable success in diverse domains, such as image classification (He et al., 2016), object detection (Redmon et al., 2016), natural language processing (Devlin et al., 2019), and federated learning (McMahan et al., 2017). Despite these advances, most current deep learning methods rely on the backpropagation (BP) algorithm (Rumelhart et al., 1986), which suffers from several limitations. First, BP is not suitable when black-box systems are embedded within the model, as it requires explicit knowledge of derivatives for every layer. Furthermore, the lack of alignment between BP and cortical learning mechanisms highlights the limited biological motivation for BP.

As an alternative to the BP algorithm, the Forward-Forward (FF) algorithm (Hinton, 2022) was proposed. Unlike BP, which typically requires one forward pass followed by one backward pass, FF performs two forward passes for each data point and updates the parameters layer-by-layer. Specifically, for each data point, the FF algorithm generates a negative sample by pairing the input with an incorrect label, and then employs a goodness metric to train each layer such that the goodness of the positive (original) pair exceeds a threshold, while that of the negative pair falls below it. The FF algorithm offers several advantages, including lower hardware requirements and adaptability to scenarios where the precise details of forward computations are unknown, since it does not require derivative propagation or storage of intermediate gradients during training. However, the quality of negative samples has a significant impact on the algorithm. To address this issue, Papachristodoulou et al. (2024) introduced the idea of grouping. Specifically, for a convolutional layer, the channels are divided into several groups, each corresponding to a single class. This structure removes the need for negative samples by enforcing a large goodness for the group associated with the target class while keeping the goodness of the remaining groups small.

Motivated by the FF algorithm and its variants, this paper proposes a forward-only algorithm for classification tasks. The proposed method is inspired by three biologically plausible observations: 1) different regions of the brain are responsible for distinct tasks, 2) neurogenesis may occur in certain

regions of the brain, influenced by learning or environmental factors, and 3) less active neurons may degenerate. Our contributions can be summarized as follows:

- Motivated by Papachristodoulou et al. (2024), we partition the output neurons into several sets, referred to as support neuron (SN) sets, with each set corresponding precisely to one class. Importantly, we generalize the grouping idea so that each SN set can include not only channels of feature maps, but also other forms of "neurons," such as selected elements of latent feature vectors in a full-connected (linear) layer. By defining the positive goodness of a data point as the squared response of its corresponding SN set, and the negative goodness as the mean squared response of the remaining sets, the algorithm enforces the positive goodness to exceed the negative goodness. This framework applies to both fully connected and convolutional layers. Building on the SN sets, we also adopt a layer-by-layer training strategy, where the training of a layer begins only after the previous layer has been fully trained. This enables flexible control over model depth: training can stop once satisfactory performance is achieved, or additional layers can be appended when necessary. Moreover, this strategy allows different layers to be trained with different loss functions.

- We introduce neuron growth and degeneration mechanisms to enable each layer to dynamically adjust its number of neurons. If the current neurons underperform, new neurons can grow to provide assistance. Conversely, if the layer performs well, inactive neurons can be removed.

- Our experiments show that the proposed algorithm outperforms other FF-based methods that do not rely on negative samples. Furthermore, training different layers with different loss functions is both practical and justified. The hybrid use of loss functions across layers yields improvements compared with training all layers using a single loss. In addition, neuron growth and degeneration further enhance the class-wise accuracy for those classes whose corresponding SN sets are modified. Finally, additional visualization experiments highlight the explainability potential of our proposed approach.

These designs not only eliminate the need for negative samples but also endow the network with flexibility in both depth and width.

## 2 RELATED WORK

**FF-based Algorithms:** While the original FF algorithm (Hinton, 2022) provides only preliminary investigations focused on fully connected layers, many subsequent works have extended this framework. Scodellaro et al. (2023) extended the FF algorithm to convolutional layers. Besides, three techniques: symmetric loss functions, batch normalization, and overlapping local updates, were proposed by Dooms et al. (2023) to enhance the performance of the FF algorithm. Moreover, while Sun et al. (2025) introduced a novel goodness design that enables training of deeper networks, Aminifar et al. (2024) developed a lightweight inference method for the FF algorithm. Additionally, Wu et al. (2024) employed learnable embeddings in the first layer to generate negative data and proposed a distance-based metric together with a layer-collaboration local update strategy. However, their reliance on negative data emphasizes a dependency on sample quality. To address this limitation, the Cascaded Forward (CaFo) framework (Zhao et al., 2025) eliminated the need for negative data by directly outputting label distributions for each cascaded block. Similarly, Papachristodoulou et al. (2024) achieved this by introducing channel-wise competitive (CwC) learning, which splits latent feature maps into groups by channels, with each group corresponding to a specific class, thereby allowing each layer to act as an independent classifier.

**Adaptive Architecture:** Traditional deep learning models are typically trained with fixed architectures, where the network structure is treated as a hyperparameter tuned on a validation set. However, the vast search space often results in unnecessary energy and time consumption. To mitigate this issue, some researchers have explored allowing models to dynamically prune unnecessary connections, thereby removing the need for retraining. Specifically, Guo et al. (2016) proposed a method that significantly reduces network complexity through on-the-fly connection pruning. Similarly, Yu et al. (2018) introduced the slimmable neural network, which allows the width of the network to be dynamically adjusted at runtime. While connection pruning is beneficial in practice, especially for edge devices with stringent energy limitations (Cai et al., 2019), it may be ineffective when the

initial architecture lacks sufficient capacity for the task. To address this limitation, Cortes et al. (2017) proposed AdaNet, an approach that adaptively learns the network structure. Nevertheless, their candidate generation process can be computationally expensive and lacks dedicated design for convolutional layers.

In this paper, we first propose the concept of support neuron (SN) sets, which extends the grouping idea in Papachristodoulou et al. (2024) to fully connected layers. Furthermore, by imitating the functioning of the brain, the neuron growth and degeneration mechanisms can be naturally designed to enable each layer to dynamically adjust the number of neurons in each set.

## 3 METHODOLOGY

### 3.1 NOTATIONS AND DEFINITIONS

For a $K$-class classification task, let $(\mathbf{x}_i, y_i)$ denote the $i$-th input–label pair in the dataset $D = \{(\mathbf{x}_i, y_i)\}_{i=1}^M$, where $M$ is the total number of samples. The input $\mathbf{x}_i$ is a real vector when the data are vector-valued, and a three-dimensional tensor when the data are images. The label $y_i \in \{1, 2, \ldots, K\}$ specifies the class of the corresponding sample.

The representation at the $l$-th layer of a neural network is defined as

$$\mathbf{z}_l = f_l(\mathbf{z}_{l-1} \mid \theta_l),$$

where the subscript $l$ denotes the layer index, $\mathbf{z}$ is the latent feature, and $\theta_l$ represents the learnable parameters of the $l$-th layer. For simplicity, we omit the sample index $i$. In particular, we have $\mathbf{z}_0 = \mathbf{x}$.

Based on these notations, we introduce the following definition.

**Definition 1** *Support Neurons (SN): Suppose the feature vector $\mathbf{z}_l$ has dimension $d_l$, i.e., the $l$-th layer of the network consists of $d_l$ neurons. Let $N_l = \{1, 2, \ldots, d_l\}$ denote the set of neuron indices in the $l$-th layer. The subset $N_l^k \subseteq N_l$ corresponds to the **support neurons** associated with class $k$, such that:*

- $\bigcup_k N_l^k = N_l$;

- $N_l^k \cap N_l^{k'} = \emptyset$ for $k \neq k'$;

- $N_l^k \neq \emptyset$.

### 3.2 EXPLAINABLE NEURAL NETWORK MODEL

In the $l$-th layer, for the $i$-th sample, if the feature $\mathbf{z}_{l,i}$ corresponds to class $k$, then the neurons in the SN set $N_l^k$ are expected to exhibit stronger responses than those in other SN sets. With this design, when an input $\mathbf{x}_i$ produces an output $\mathbf{z}_{L,i}$ through an $L$-layer neural network, the SN set with the strongest response determines the predicted class.

Formally, let $\mathbf{z}_l$ be the latent feature of an arbitrary input $\mathbf{x}$. We define

$$g_l^k = \sum_{j \in N_l^k} (z_l(j))^2$$

as the response level (or goodness level) of the SN set $N_l^k$, where $z_l(j)$ denotes the $j$-th component of $\mathbf{z}_l$. For an $L$-layer network, the predicted class of a given input can be determined in two ways:

$$\text{Last Layer Response (LLR):} \quad y' = \arg\max_k g_L^k,$$

$$\text{Overall Layer Response (OLR):} \quad y' = \arg\max_k \sum_{l=1}^L g_l^k.$$

This mechanism draws inspiration from the brain, where specific types of stimuli activate distinct regions, analogous to the activation of the corresponding SN sets.

In this paper, the neurons in the $l$-th layer are initially uniformly partitioned, and the labels are assigned to the sets in a sequential manner.

### 3.3 Layer Design

**Linear Layer:** The design of the linear layer is straightforward: the output neurons are partitioned into $K$ sets, with each set dedicated to responding to a particular class. The response level is computed as

$$g_l^k = \sum_{j \in N_l^k} (z_l(j))^2.$$

**Convolutional Layer:** The design of the convolutional layer is more subtle. A naive approach is to flatten the output feature maps into a vector and then construct the SN sets and response levels in the same manner as in a linear layer. However, this may lead to neurons from different SN sets residing in the same channel, which prevents the corresponding convolutional kernels from learning class-specific representations effectively, potentially resulting in suboptimal performance.

A more principled approach is to partition the feature maps along the channel dimension, as proposed in Papachristodoulou et al. (2024). Suppose the output of the $l$-th layer is $z_l \in \mathbb{R}^{C_l \times W_l \times H_l}$, where $C_l$, $W_l$, and $H_l$ denote the number of channels, width, and height of the feature maps, respectively. Each channel is regarded as a neuron, and the set of channels is divided into SN sets, i.e., $N_l^k \subseteq \{1, 2, \ldots, C_l\}$. In this formulation, each SN set corresponds to a group of channels rather than individual vector elements. The response level of $N_l^k$ is then computed by summing the squared activations across all pixels within the channels of the set.

**Remark 1** *The design of the convolutional layer is closely related to the approach in Papachristodoulou et al. (2024), with distinctions and further discussion provided in Section 3.7.*

### 3.4 Training

Following the FF algorithm (Hinton, 2022), the proposed method is trained layer by layer. Specifically, the first layer $f_1(\mathbf{x} \mid \theta_1)$ is trained on the original training set, and each subsequent layer is trained on the dataset formed by the outputs of the preceding layer. The overall training procedure is summarized in Appendix A due to the space limitation. This layer-wise training strategy provides flexibility in network depth: if a particular layer achieves satisfactory performance for the task, the remaining layers may be skipped or their training terminated.

We consider three types of loss functions for training:

- **Positive vs. Negative (PvN) loss:** Adapted from the original FF algorithm (Hinton, 2022), this loss is proposed by Papachristodoulou et al. (2024). It enforces the response of the correct SN set to exceed a threshold $\phi$, while the responses of the other SN sets remain below that threshold. It is formulated as:

$$L_{PvN} = \frac{1}{M} \sum_{i=1}^{M} \left[ \log \left(1 + \exp(-g_{l,i}^y + \phi)\right) + \log \left(1 + \exp \left(\frac{1}{K-1} \sum_{k \neq y} g_{l,i}^k - \phi\right)\right)\right].$$

- **SN-set-wise loss (CwC loss):** Inspired by the CwC loss from Papachristodoulou et al. (2024), this objective maximizes the proportion of the response contributed by the correct SN set among all SN sets. It is given by:

$$L_{CwC} = \frac{1}{M} \sum_{i=1}^{M} \log \left(\frac{\exp(g_{l,i}^y)}{\sum_{k=1}^{K} \exp(g_{l,i}^k)}\right).$$

- **Cross-Entropy (CE) loss:** By concatenating the responses of all SN sets into a vector $\mathbf{g}_{l,i} = [g_{l,i}^1, g_{l,i}^2, \ldots, g_{l,i}^K]^T$. This $K$-dimensional vector can be used to obtain a probability distribution. The standard CE loss can therefore be applied:

$$L_{CE} = \frac{1}{M} \sum_{i=1}^{M} CE(\mathbf{g}_{l,i}, y).$$

Notably, this formulation is identical to the $L_{CWC}$ when applying the Softmax function to calculate the probability distribution. However, alternative functions can also be used to obtain the probability distribution.

## 3.5 NEURON GROWTH

In the human brain, when a task is too complex for the existing neurons in a specific region to handle effectively, new neurons may grow to assist in accomplishing the task.

Inspired by this mechanism, our model allows new neurons to grow when an SN set fails to adequately recognize data from its corresponding class. The growth of neurons is designed through the following steps:

1) **Decision to grow:** For layer $l$, if the class-wise accuracy $acc_k$ (i.e., the accuracy on samples belonging to class $k$) falls below a given threshold, then a new set of neurons

$$N_l^{k,*} = \{d_l + 1, d_l + 2, \ldots, d_l + d_l'\}$$

is added to support the task, where $d_l'$ is the number of newly introduced neurons.

2) **Neuron initialization:** The weights of the new neurons (a weight matrix in the linear case or a set of kernels in the convolutional case) are initialized. Initialization can be random, or based on copying the neuron that exhibits the highest average response to samples in class $k$. This is achieved by evaluating all samples of class $k$ and recording the average responses of the original neurons.

3) **Neuron training:** Once initialized, the new neurons are trained on the class $k$ samples misclassified by layer $l$. However, it should be noted that the training algorithm for the new neurons should be subtle, as the representational capacity of a small number of weights is limited, particularly when random initialization is used.

In this work, we propose a simple yet effective algorithm for neuron growth, which combines copy-based initialization with a tailored training strategy:

1) Collect the misclassified data $X_{mis}$ from class $k$, together with an equal number of samples $X_{dif}$ from other classes. Construct a new dataset

$$D_{new} = \{X_{mis}, X_{dif}\} \times \{Y_{mis}, Y_{dif}\},$$

where $\times$ denotes the Cartesian product.

2) For each batch of $D_{new}$, compute the response $g_{l,i}^{k,*}$ of the new neuron set $N_l^{k,*}$ for each input.

3) Define the loss function as

$$L = \sum_{i:\mathbf{x}_i \in X_{mis}} g_{l,i}^{k,*} - \sum_{i:\mathbf{x}_i \in X_{dif}} g_{l,i}^{k,*},$$

which encourages the new neurons to respond strongly to misclassified samples while remaining inactive for inputs not belonging to their SN set.

## 3.6 NEURON DEGENERATION

It is also well known that in the human brain, neurons that remain inactive regardless of input stimuli may eventually degenerate. Inspired by this phenomenon, we also incorporate neuron degeneration into our framework. The procedure is straightforward: assume a neuron belongs to the SN set $N_l^k$. If its average response across all samples in class $k$ is less than or equal to a given threshold, the neuron is removed by eliminating its corresponding weights. Notably, the degeneration threshold can be chosen proportional to the threshold used in the PvN loss when applicable.

By combining neuron growth and degeneration, the network gains flexibility not only in depth but also in width. When performance is already sufficient, inactive neurons can be pruned. Conversely, if additional representational capacity is required, new neurons can be introduced.

**Remark 2** *It is important to consider the effect of neuron growth and degeneration on subsequent layers. However, the trade-off between the adjusted layer and its successors may introduce new challenges. A natural way to address this is to adopt a progressive, layer-by-layer procedure: first train the initial layer, apply neuron growth and degeneration if necessary, and then proceed sequentially to the following layers. Nevertheless, this paper focuses only on a one-layer (last-layer) investigation as a preliminary study, with the aim of providing useful insights.*

### 3.7 RELATION TO EXISTING METHODS

Our approach bears connections to existing work while introducing key distinctions:

- **Comparison with double forward methods (Hinton, 2022):** Similar to these methods, our approach employs the concept of goodness (or response level). However, in their frameworks, each data point must be processed twice during training, whereas in our method, only a single forward pass is required, enabled by the SN design.

- **Comparison with Papachristodoulou et al. (2024):**

  While the convolutional layer design is similar, there are several key differences.

  First, the training methodology differs. Their approach trains each layer sequentially using mini-batches, whereas our method trains each layer directly on the entire dataset. This distinction offers several advantages: 1) The optimization in our method is essentially closed-form, relying on stochastic gradient algorithms applied independently at each layer without inter-layer dependencies. This potentially simplifies both theoretical analysis and the development of new optimization algorithms. 2) The combination of our training scheme with the SN set concept provides the model with greater flexibility in both depth and width. In terms of width, the architecture can evolve automatically through neuron growth and degeneration. In terms of depth, training can stop once a layer achieves satisfactory performance on the training set, while additional layers can still be added without re-training previous ones if further performance improvement is needed.

  Second, when the definition of "neurons" changes, the computation of goodness (or responses) can also vary, for example, defining a "neuron" as a single element in the feature vector or a single pixel in the feature maps. However, for convolutional layers, our empirical tests show that channel-wise partitioning yields the best performance.

## 4 EXPERIMENTS

### 4.1 SETTINGS

**Dataset:** We evaluate our method on several benchmark datasets: MNIST (LeCun et al., 1998), CIFAR-10, and CIFAR-100 (Krizhevsky et al., 2009). All reported results are based on the test sets.

**Comparing algorithms:** For FF and its variants, we include the original FF algorithm (Hinton, 2022), its convolutional extension (FFconv) (Enan, 2025), DeeperForward (Sun et al., 2025), CaFo (Zhao et al., 2025), and CFSE (Papachristodoulou et al., 2024). We also compare against additional non-BP and biologically plausible methods, including DRTP and BWBPF. Moreover, ResNet18 (He et al., 2016) and deep neural networks with the same architecture as our method are employed as baseline models.

**Network structure:** All algorithms are implemented using PyTorch (Paszke et al., 2019). The network architectures are selected according to the dataset and algorithm: fully connected layers are employed for MNIST, while convolutional neural networks followed by a linear layer are adopted for CIFAR-10 and CIFAR-100. For MNIST, only linear networks are used to specifically verify the effectiveness of linear layers, while the effectiveness of convolutional layers is examined using the CIFAR datasets. In particular, we allocate 2 neurons per class in the final linear layer for CIFAR-10 and 4 neurons per class for CIFAR-100. Since the initial layers in our method achieve relatively low accuracies, we adopt the LLR criterion to obtain predictions. Detailed architectures and hyperparameter settings are provided in the Appendix.

**Devices:** Experiments were conducted on a system equipped with an Intel(R) Xeon(R) Platinum 8383C CPU (2.70GHz, 40 cores, 160 logical processors) and an NVIDIA GeForce RTX 4090 GPU (driver version 550.54.14) with 24,564 MiB of available GPU memory.

## 4.2 EFFECTS OF THE LOSS FUNCTIONS

We begin by analyzing the effect of different loss functions on the training process of our algorithm. Notably, although the CWC loss is equivalent to the CE loss with Softmax, in our implementation we adopt the clamping technique from Papachristodoulou et al. (2024) to prevent excessively large goodness values and stabilize the training process for the CWC loss. For the CE loss, we use the standard implementation provided by the PyTorch package (Paszke et al., 2019). Additionally, we implement the CE loss with the Sparsemax function, referred to as Sparse CE. The results are presented in Table 1, with the best performance highlighted in bold. We use *Highest* and *Last* to

Table 1: Accuracy of different loss functions

| Methods | MNIST | CIFAR-10 | CIFAR-100 |
|---|---|---|---|
| Ours_CE(Highest) | 68.5% ± 8.3% | 79.5% ± 0.5% | 51.3% ± 0.4% |
| Ours_CWC(Highest) | 70.3% ± 5.8% | 80.2% ± 0.5% | 46.6% ± 0.3% |
| Ours_PvN(Highest) | **96.6% ± 0.1%** | 72.3% ± 0.4% | 41.5% ± 0.4% |
| Ours_Sparse CE(Highest) | 88.3% ± 2.6% | 50.0% ± 7.9% | 5.5% ± 2.6% |
| Ours_CE(Last) | 39.7% ± 14.3% | 79.0% ± 0.5% | 46.8% ± 1.2% |
| Ours_CWC(Last) | 44.2% ± 16.2% | 10.0% ± 0.0% | 1.0% ± 0.0% |
| Ours_PvN(Last) | **96.6% ± 0.1%** | 72.3% ± 0.4% | 40.8% ± 0.2% |
| Ours_Sparse CE(Last) | 82.9% ± 6.2% | 50.0% ± 7.9% | 5.5% ± 2.6% |
| Ours_Combo(Highest) | Equal to PvN | **80.7% ± 0.4%** | **52.0% ± 0.3%** |
| Ours_Combo(Last) | Equal to PvN | **80.7% ± 0.4%** | **52.0% ± 0.3%** |

distinguish between the accuracy of the layer that achieved the best performance and that of the final layer. For a two-layer network, the two values are identical if the last layer's accuracy exceeds that of the first layer; otherwise, *Highest* corresponds to the accuracy of the first layer.

Our experiments show that the highest accuracy is sometimes achieved by the last layer, but it may be otherwise. For example, in the setting of CWC and CIFAR-10, the highest accuracy is achieved by an internal layer. Moreover, PvN proves effective for training linear layers but is less suitable for convolutional layers, as it achieves satisfactory performance on the MNIST dataset but suffers from suboptimal results with convolutional networks. Conversely, CwC is well-suited for convolutional layers but fails to train linear layers, as the *Highest* accuracies on both CIFAR datasets are observed in hidden convolutional layers, while the final linear layers perform poorly. Interestingly, although CE performs poorly on MNIST, it performs relatively well on both CIFAR datasets, where the *Highest* consistently equals the *Last*, highlighting its suitability for both linear and convolutional layers. This phenomenon suggests that CE may be more appropriate for complex models or tasks but less effective in simpler ones. Moreover, the sparse CE seems to be unsuitable for all layers.

As aforementioned, since our model is trained layer-by-layer, employing different loss functions to different layers is possible. Thus, based on these observations, we further adopt a hybrid strategy: convolutional layers are trained with CWC, while the linear layer is trained with PvN. We refer to this scheme as *Combo*. Experimental results demonstrate that Combo consistently outperforms configurations trained with a single loss function, and the *Highest* accuracy is always obtained by the last linear layer, thereby reinforcing our conclusions regarding the complementary strengths of PvN and CWC.

## 4.3 PERFORMANCE ANALYSIS AND EVALUATION

After establishing the network structure and corresponding loss functions, we compare our method against existing approaches. The results are summarized in Table 2, where the best results obtained by BP-based methods are highlighted in bold, and those achieved by FF-based methods are under-lined. For our method, the *Combo* strategy is employed for training.

We aim to separately evaluate the effectiveness of linear layers and convolutional layers on the MNIST and CIFAR datasets, respectively. Since MNIST is a relatively simple dataset on which convolutional layers do not provide a substantial accuracy improvement over linear layers, forcing all methods to use a two-layer linear network allows us to fairly assess the intrinsic learning ability of linear layers without affecting the final performance comparison. In contrast, for the CIFAR datasets, linear layers alone are insufficient to obtain meaningful performance, so we adopt convolutional architectures for all methods.

For the FF algorithm, we use its original implementation on MNIST and employ FFconv with the same architecture as ours on the CIFAR datasets. For CFSE, we keep the network architecture identical to ours and report results under two prediction schemes: (i) goodness-based classification (Gd), which uses the OLR method, and (ii) softmax-based classification (Sf), which adds a fully connected classifier followed by a softmax. For CaFo, we report results from both its original architecture (Or) and our re-implemented version (Re). For both CaFo and CFSE, we report the best performance among the available choices. Note that because we only use linear layers on MNIST, the architecture of CaFo on MNIST corresponds to the Re version. For DeeperForward and BWBPF, the architectures follow ours for both MNIST and CIFAR. For DRTP, we retain its original convolutional architecture to preserve its reported performance characteristics.

Additionally, training hyperparameters, including but not limit to learning rate, optimizer, and epoch, are kept consistent across all methods.

Table 2: Accuracy of different algorithms

| Methods | MNIST | CIFAR-10 | CIFAR-100 |
|---|---|---|---|
| ResNet | Not Linear | 79.3% ± 0.3% | 55.2% ± 0.8% |
| DNN | **97.9% ± 0.2%** | **81.0% ± 1.1** % | **65.4% ± 0.3%** |
| DRTP | 89.4% ± 1.6% | 57.0% ± 1.3% | 26.7% ± 0.9% |
| BWBPF | 92.5% ± 2.3% | 80.7% ± 0.2% | 54.3% ± 1.0% |
| FF(conv) | 91.6% ± 0.4% | 67.9% ± 0.5% | 49.1% ± 0.6% |
| DeeperForward | 91.3% ± 1.4% | 64.3% ± 0.7% | 30.8% ± 0.9% |
| CaFo | 81.8% ± 0.7% (Re) | 64.8% ± 0.7% (Or) | 40.1% ± 0.4% (Re) |
| CFSE_CwC | Not Linear | 77.2% ± 0.6% (Sf) | 48.9% ± 0.6% (Gd) |
| Ours | 96.6% ± 0.2% | 80.5% ± 0.4% | 52.0% ± 0.3% |

The results demonstrate that our method consistently outperforms other FF-based algorithms while achieving accuracies that are competitive with other non-BP methods and certain BP-based training approaches. Furthermore, compared with all non-BP and FF-based methods, our approach exhibits superior adaptivity on fully connected layers.

## 4.4 Neuron Growth and Degeneration

In this section, we evaluate the performance of the network after applying neuron growth and degeneration. For MNIST, we employed a one-layer linear architecture of shape $784 \rightarrow 50$, trained for only one epoch. This limited training ensures that the network does not reach its performance bottleneck, leaving room for accuracy improvement. For CIFAR10, we adopted a one-layer convolutional architecture with 20 output channels, kernel size 3, stride 1, and padding 1. The network was trained for 100 epochs, allowing it to reach its performance bottleneck. This setup enables us to investigate whether growth and degeneration can further enhance performance. We denote the network after initial training as $net$.

**Growth:** We grew 1 and 2 new neurons (or channels) in the SN set with the lowest class-wise accuracy for MNIST and CIFAR10, respectively. Let $net_{clone}$ represent the network that only uses clone initialization, and $net_{train}$ the network that undergoes the full training procedure based on $net$. The training durations for $net_{train}$ were set to 5 epochs for MNIST and 40 epochs for CIFAR-10, with learning rates fixed at 0.0001 and 0.00001, respectively.

The results are reported in Table 3, where "Acc." denotes the overall accuracy and "Acc. $k$" the accuracy for class $k$. The class in which a new neuron was added is highlighted in bold.

Table 3: Accuracy after growth.

| | MNIST | | | | | |
| --- | --- | --- | --- | --- | --- | --- |
| Net | Acc. | Acc. 1 | Acc. 2 | Acc. 3 | Acc. 4 | Acc. 5 |
| $net$ | 76.95 | 95.82 | 94.98 | 87.89 | **6.73** | 89.82 |
| $net_{clone}$ | 82.09 | 95.82 | 94.98 | 87.79 | **58.31** | 89.82 |
| $net_{train}$ | 83.01 | 95.82 | 94.98 | 87.69 | **69.11** | 89.82 |
| Net | Acc. | Acc. 6 | Acc. 7 | Acc. 8 | Acc. 9 | Acc. 10 |
| $net$ | 76.95 | 79.93 | 90.71 | 93.19 | 81.11 | 48.66 |
| $net_{clone}$ | 82.09 | 79.26 | 90.71 | 93.19 | 81.11 | 48.66 |
| $net_{train}$ | 83.01 | 77.80 | 90.71 | 93.19 | 80.80 | 48.66 |
| | CIFAR10 | | | | | |
| Net | Acc. | Acc. 1 | Acc. 2 | Acc. 3 | Acc. 4 | Acc. 5 |
| $net$ | 36.88 | 42.80 | 42.80 | **12.70** | 26.30 | 43.30 |
| $net_{clone}$ | 36.88 | 42.80 | 42.80 | **12.70** | 26.30 | 43.30 |
| $net_{train}$ | 37.04 | 42.70 | 42.80 | **23.10** | 26.60 | 35.20 |
| Net | Acc. | Acc. 6 | Acc. 7 | Acc. 8 | Acc. 9 | Acc. 10 |
| $net$ | 36.88 | 31.10 | 52.20 | 27.60 | 52.50 | 37.50 |
| $net_{clone}$ | 36.88 | 31.10 | 52.20 | 27.60 | 52.50 | 37.50 |
| $net_{train}$ | 37.04 | 30.90 | 51.40 | 27.60 | 52.60 | 37.50 |

As observed, when the task is relatively simple (MNIST), both clone initialization and full training significantly improve the accuracy of the targeted class, since the cloned neuron can effectively respond to the correct samples. Nevertheless, this may slightly decrease the accuracy of other classes. Importantly, the overall accuracy still increases. In contrast, for the more challenging task (CIFAR10), the neuron with the highest response was not strong enough due to the limited representational capacity of the original model. Consequently, clone initialization alone yielded little improvement. However, the full training procedure still enhanced the performance of the $k$-th class and the overall accuracy, albeit with some reduction in other classes. This trade-off likely arises because the new neurons occasionally responded incorrectly to samples from the other classes.

**Degeneration:** For both MNIST and CIFAR10, we deleted two neurons in sequence. To clearly demonstrate the effects of degeneration, we did not apply a threshold-based rule; instead, at each step we removed the neuron with the lowest response across all neurons. The results are presented in Table 4, where "Num. of Del." denotes the number of deleted neurons, and the class from which neurons were removed is highlighted in bold.

The results reveal a phenomenon similar to that of growth: the accuracy of the class from which neurons were removed improved, but at the expense of decreased accuracy in other classes. Importantly, while moderate degeneration does not substantially affect overall accuracy, excessive degeneration can be harmful. This occurs because deleted neurons not only reduce the average response level of their own SN sets (thereby lowering the accuracy of their corresponding classes before degeneration), but also contribute to suppressing responses for other classes, which enables the correct SN sets to respond more strongly than the degenerated ones due to the existence of the inactive neurons. In practice, a well-trained layer with sufficiently good performance often contains redundant neurons, which can be safely pruned through degeneration.

Notably, the results in this section are based on a single run, as the weakest class may vary across runs and averaging could obscure subtle improvements. To enhance robustness, we provide additional runs in the Appendix.

**Remark 3** *Growing a network from scratch is indeed feasible, and we have conducted some preliminary experiments in this direction. However, since growth and degeneration did not yield sub-*

Table 4: Accuracy after degeneration.

| | MNIST | | | | | |
|---|---|---|---|---|---|---|
| Num. of Del. | Acc. | Acc. 1 | Acc. 2 | Acc. 3 | Acc. 4 | Acc. 5 |
| 0 | 79.20 | 90.51 | 91.98 | 76.64 | 75.64 | 63.95 |
| 1 | 79.08 | **98.06** | 91.98 | 74.32 | 74.85 | 63.74 |
| 2 | 79.36 | 97.75 | 91.89 | **88.85** | 68.71 | 63.03 |
| Num. of Del. | Acc. | Acc. 6 | Acc. 7 | Acc. 8 | Acc. 9 | Acc. 10 |
| 0 | 79.20 | 81.61 | 93.11 | 40.27 | 85.83 | 92.76 |
| 1 | 79.08 | 78.69 | 91.64 | 39.88 | 85.42 | 92.36 |
| 2 | 79.36 | 78.25 | 89.97 | 39.01 | 83.88 | 91.87 |
| | CIFAR10 | | | | | |
| Net | Acc. | Acc. 1 | Acc. 2 | Acc. 3 | Acc. 4 | Acc. 5 |
| 0 | 38.37 | 39.00 | 48.60 | 21.20 | 31.00 | 38.80 |
| 1 | 20.12 | 23.00 | 13.50 | 6.80 | 0.20 | 13.30 |
| 2 | 19.16 | 21.30 | 6.60 | **62.60** | 0.00 | 0.00 |
| Num. of Del. | Acc. | Acc. 6 | Acc. 7 | Acc. 8 | Acc. 9 | Acc. 10 |
| 0 | 38.37 | 31.40 | 49.50 | 29.30 | 55.90 | 39.00 |
| 1 | 20.12 | **97.40** | 4.10 | 0.30 | 38.70 | 3.90 |
| 2 | 19.16 | 71.60 | 0.00 | 0.00 | 26.90 | 2.60 |

*stantial improvements in overall accuracy, no specific strategy consistently led to significant gains. Nevertheless, we believe that the idea of growth and degeneration holds promise and may inspire further investigations into dynamic network adjustment.*

## 5 CONCLUSION AND PERSPECTIVES

This paper introduced the concept of SN sets to extend the grouping idea to linear layers and further proposed neuron growth and degeneration mechanisms inspired by the working principles of the brain. Experiments demonstrated the superior performance of the proposed algorithm and validated some positive effects of neuron growth and degeneration, while also showcasing some limitations.

The current study of neuron growth and degeneration is still at an early stage, as only one-layer architectures have been investigated. In practice, inserting and deleting neurons within deeper internal layers may also occur; hence, this direction warrants future exploration. Moreover, the assignment of neurons to SN sets remains underexplored. Investigating strategies for neuron selection, as well as the potential effects of overlapping neurons across SN sets, could lead to novel formulations involving combinatorial optimization problems. Finally, unlike the dense connections in our model, neurons in the brain typically connect only with a limited set of neighbors. Introducing connection sparsity could therefore be a promising future direction, making the model both closer to biological neural systems and more storage-efficient.

## 6 ETHICS STATEMENT

This work complies with the ICLR Code of Ethics. Our study focuses on developing a novel learning paradigm, Forward-Forward Learning with dynamic architecture adaptation, for classification tasks. The research does not involve human subjects, sensitive personal data, or privacy-related datasets. We only use publicly available benchmark datasets that are widely adopted in the community, ensuring compliance with licensing terms and ethical standards. We have also considered the potential societal impact of our method: while improved classification efficiency can enable broader applications, our contribution is intended for scientific and educational purposes, and we have taken care to avoid promoting misuse in sensitive domains.

## 7 REPRODUCIBILITY STATEMENT

We have made every effort to ensure the reproducibility of our results. The paper includes a complete description of the proposed algorithm, the architecture adaptation mechanism, and the training and evaluation procedures. Hyperparameter settings, implementation details, and experimental protocols are provided in the appendix and supplementary materials. All datasets used in our experiments are standard public benchmarks, and preprocessing steps are documented. We will also release anonymous source code as supplementary material to facilitate independent verification and reuse of our work.

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

---

**Algorithm 1** Training algorithm.

---

**Input**: Training set $D = \{(\mathbf{x}_i, y_i)\}_{i=1}^{M}$, $L$-layers model $\{f_l(\cdot|\theta_l)\}_{l=1}^{L}$, maximal epoch $E$.
**Output**: Trained model.

  1: Let $e = 0, l = 0$;
  2: Let $Z_0 = \{\mathbf{x}_i\}_{i=1}^{M}$;
  3: Let $Y = \{y_i\}_{i=1}^{M}$;
  4: Let $D_0 = Z_0 \times Y$;
  5: **for** $l = 1$ to $L$ **do**
  6:     Train $f(\cdot \mid \theta_l)$ using the loss function $L_l$ for this layer;
  7:     $Z_l = f(Z_{l-1}|\theta_l)$;
  8:     $D_l = Z_l \times Y$;
  9: **end for**

---

## A    DETAILED NETWORK ARCHITECTURES

For MNIST, we construct a purely linear network consisting of two fully connected layers, i.e., $Linear(784, 100) \rightarrow Linear(100, 100)$, each followed by a ReLU activation. For BP-based methods, an additional classifier $Linear(100, 10)$ is appended to produce the final prediction.

For CIFAR-10 and CIFAR-100, the network architecture is built from a sequence of convolutional blocks. We define a block, denoted as `Block(in, out, Pool)`, which consists of a convolutional layer, a ReLU activation, a batch normalization layer, and optionally a max-pooling layer. Here, `in` and `out` denote the input and output channel dimensions, respectively, and `Pool` is a Boolean flag indicating whether max-pooling is applied. Each convolutional layer uses a kernel size of 3, stride of 1, and padding of 1. Whenever max-pooling is enabled, it is applied with a kernel size of 2 and stride of 2.

- **CIFAR-10:** The network structure is:
  $$Block(3, 20, \text{False}) \rightarrow Block(20, 80, \text{True}) \rightarrow Block(80, 240, \text{False}) \rightarrow$$
  $$Block(240, 480, \text{True}) \rightarrow Linear(-1, 20),$$

  where $-1$ indicates that the input dimension is adaptively determined by the output of the previous block.

- **CIFAR-100:** The network structure is
  $$Block(3, 60, \text{False}) \rightarrow Block(60, 120, \text{True}) \rightarrow Block(120, 240, \text{False}) \rightarrow$$
  $$Block(240, 400, \text{True}) \rightarrow Block(400, 800, \text{False}) \rightarrow Block(800, 1600, \text{True}) \rightarrow$$
  $$Linear(-1, 400).$$

  For our method, the first four blocks are trained on the 20 super-labels, while the remaining layers are trained on the full set of class labels.

In the CIFAR experiments, the convolutional block structures are kept identical for CFSE and CaFo(re), and the main differences lie in their prediction strategies, i.e., the last linear layer.

For both MNIST and CIFAR experiments, the learning rate is fixed at $1 \times 10^{-3}$, and the threshold for the PvN loss is set to 2.0. The batch size and number of training epochs are set to 128 and 100, respectively, across all methods. Moreover, we do not apply any data preprocessing or augmentation techniques, using only normalization, in order to demonstrate the performance under the simplest settings.

Here we also provide the pseudo-code for our algorithm to facilitate a better understanding of our training process as shown in Algorithms. 1 and 2.

## B    FUTURE DIRECTIONS

As discussed in Section 5, several directions warrant further investigation, including neuron growth and degeneration in hidden layers, the selection of neurons, overlapping neurons between SN sets, and sparse connectivity. Beyond these, there are also additional avenues for future research.

---

**Algorithm 2** Layer-wise training algorithm.

---

**Input**: Training set for layer $l$ $D_l$, the $l$-th layer $f_l(\cdot|\theta_l)$, maximal epoch $E$.
**Output**: Trained layer.

1: Let $e = 0$;
2: **for** $e = 1$ to $E$ **do**
3:    **for** each random batch of $D_l$ **do**
4:       Train $f(\cdot \mid \theta_l)$ with the optimizer;
5:    **end for**
6: **end for**

---

### B.1 RELATION WITH HIERARCHICAL CLUSTERING

In our CIFAR-100 experiments, the first several layers were not trained on the original labels but instead on the 20 super-classes of CIFAR-100, demonstrating the feasibility of this training strategy. However, in most datasets, such human-crafted super-classes may not exist. A natural solution is to employ hierarchical clustering to automatically generate super-labels.

For a simple illustration, as shown in Fig. 1, consider three categories: white cat, black cat, and dog. A hierarchical clustering algorithm may construct a binary tree whose first split separates "cat" from "dog," where "cat" is not pre-defined but rather discovered by the algorithm. In this case, the first block of the network can be trained using the super-labels at the first level (cat vs. dog), while the subsequent block can be trained on the leaf-level labels (white cat, black cat, dog).

More generally, suppose we have $K$ classes and a hierarchical clustering algorithm $\mathcal{A}$ that, given a training set $D$, produces a binary tree $\mathbf{T} = \mathcal{A}(D)$. For the $l$-th layer of the tree, there will be $O(2^l)$ super-labels. If an internal node at layer $l$ corresponds to a leaf class, it is also treated as a label in subsequent layers. The algorithm $\mathcal{A}$ must ensure that the tree $\mathbf{T}$ has exactly $K$ leaf nodes, each corresponding to one of the original classes in $D$. With this tree, the network architecture can be designed by aligning each block with a layer of the tree and training it based on the corresponding super-labels.

This method can potentially enhance the scalability of our method to complicated tasks. However, the main challenge lies in the design of such an clustering algorithm.

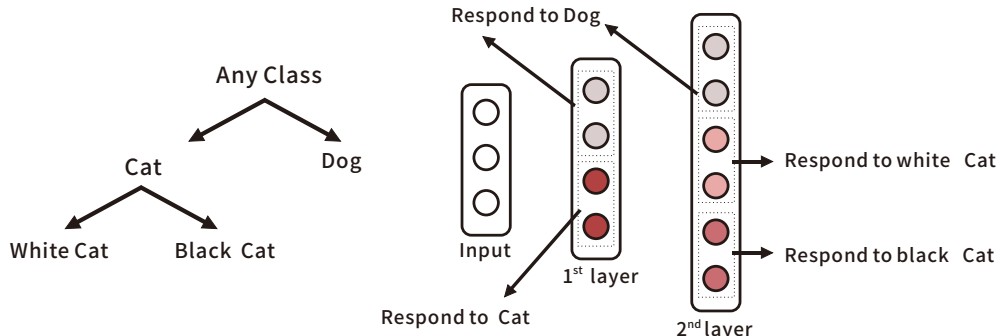

Figure 1: Simple Example for Hierarchical clustering and the proposed method.

### B.2 BRAIN NETWORK

As mentioned in Remark 2, a network can grow layer by layer. However, this paper provides only limited insights, and several open questions remain. For instance, should the entire network be grown from scratch, or should growth begin from an initial model? For the former, the challenges include (but are not limited to): how to design the growth paradigm, how to determine the number of neurons for layer $l$, how many neurons should be added at each step, when growth should stop, and how degeneration should be carried out. For the latter, the central issue lies in deciding the

initial model. The aforementioned hierarchical clustering approach may provide a potential solution. Furthermore, once an initial model is fixed, neuron growth within internal layers becomes inevitable, which raises additional challenges that warrant further investigation.

## C  THE USE OF LARGE LANGUAGE MODELS (LLMS)

In this work, large language models (LLMs) were employed to refine the writing style of the manuscript, to translate pseudo-code developed by the authors into executable code, and to facilitate the search for related literature (e.g. Guo et al. (2016), Yu et al. (2018), (Cai et al., 2019) and Cortes et al. (2017)). Beyond these supportive roles, LLMs did not contribute to the conceptual development, experimental design, or results of the study.

## D  EXTRA RESULTS

### D.1  LATENT ACTIVATIONS

In this section, we visualize the activations of neurons across layers. For a trained neural network, we sequentially feed samples from each class and record the class-wise average responses of all neurons. For MNIST and CIFAR-10, we use all 10 classes. For CIFAR-100, we select four classes from the same superclass (e.g., bottle, bowl, can, and cup) and one class from a different superclass (e.g., apple) to better analyze cross-class behavior.

The results for MNIST and CIFAR-10 are shown in Fig. 2. As illustrated, in MNIST, the responses of neurons in the correct SN set are consistently and substantially higher than those in other sets, clearly reflecting class-specific specialization. In contrast, when the data become more complex (CIFAR-10), the distinction between the correct SN set and the others becomes less pronounced in the first two layers. Nevertheless, a small number of strong neurons in the correct SN set gradually emerge in layers 3 and 4. In the final linear layer, the response of the correct SN set ultimately surpasses the others, although the margin is relatively less significant compared with MNIST.

The results for CIFAR-100 are presented in Fig. 3. The overall pattern resembles that observed in CIFAR-10; however, the network struggles to distinguish among the classes bowl, cup, and can. Within this superclass, only the class bottle exhibits a clear dominance of the correct SN set in the final layers, similar to the class apple. Interestingly, in the first four layers, the neurons display nearly identical behaviors for bottle, bowl, can, and cup. This is a direct consequence of the training procedure, as these layers are trained using superclass labels. Accordingly, activations for apple, which comes from a different superclass, differ substantially in these layers. In the final three layers, neuron responses begin to diverge across classes. Moreover, as the same in CIFAR-10, several prominent neurons emerge in the deeper layers for classes such as bottle and apple.

### D.2  FURTHER INVESTIGATION ON GROWTH AND DEGENERATION

#### D.2.1  GROWTH

We conduct additional experiments to illustrate how neuron growth influences the neuron responses of neural networks. We first focus on the first layer of the MNIST and CIFAR10 networks, both of which are well trained under the default settings, allowing the growth effects to be more clearly observed. Following the procedure in Section D.1, we input data class-wise to visualize class-wise responses. For both datasets, we grow two neurons simultaneously for better visualization. The hyperparameters of growth follow those described in Section 4.4.

The responses are shown in Figs. 8a and 4b. In the figures, the pink and brown bars denote the correct and incorrect SN sets, respectively, consistent with the previous sections. The blue and red bars represent the responses of the newly added neurons, where blue indicates that the new neurons are not part of the current SN set of the pink neurons, and red indicates the opposite.

As shown in Fig. 8a, the newly grown neurons on MNIST are able to respond appropriately to both the correct and incorrect SN sets: their activations are strong when the inputs belong to their SN sets, and weak otherwise. However, for CIFAR10 (Fig. 4b), such behavior is less evident. This may

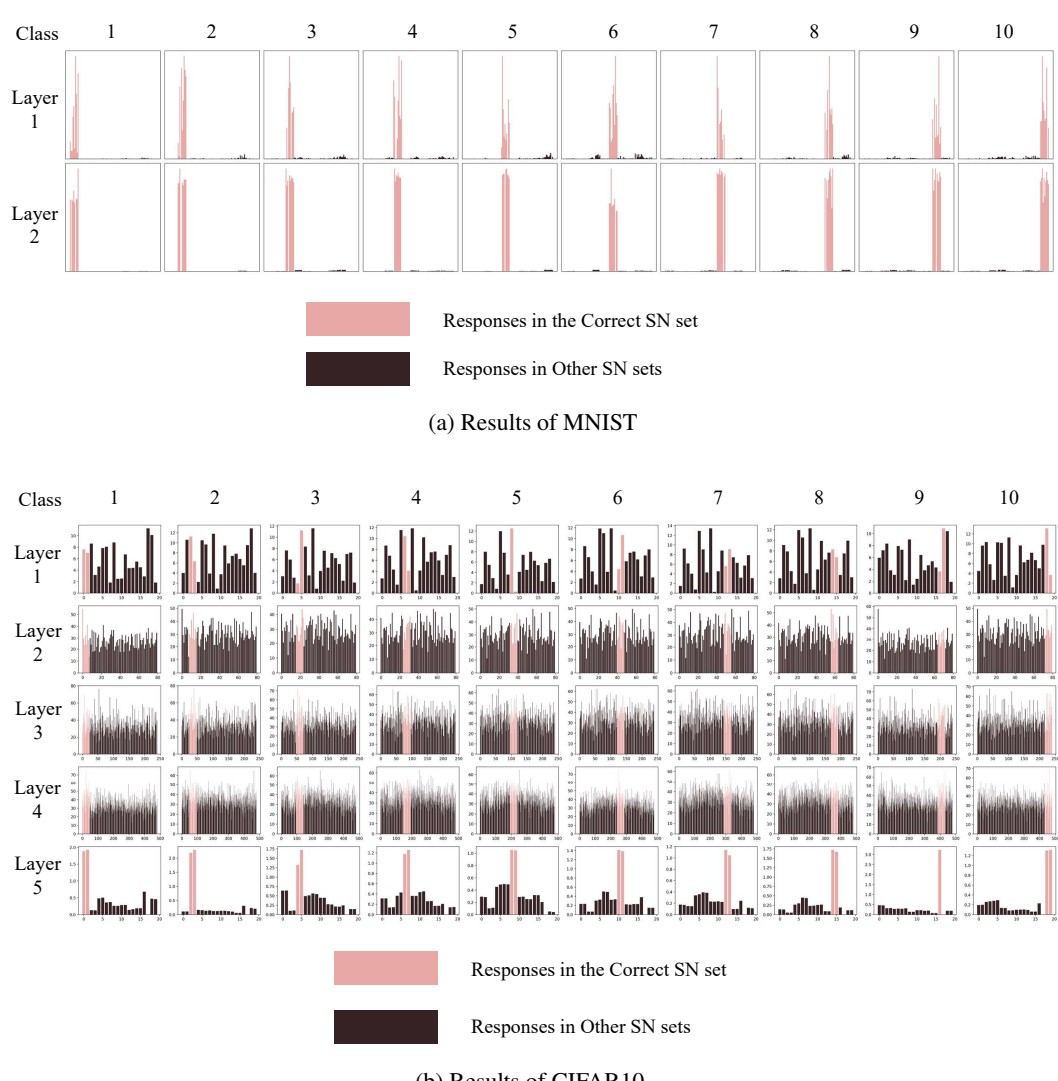

(a) Results of MNIST

(b) Results of CIFAR10.

Figure 2: Neurons' responses across layers. In each subfigure, the x-axis denotes the neuron indices, and the bars represent their corresponding response values.

be attributed to the fact that the representational capacity of the newly added neurons is strongly dependent on the existing neurons due to the clone-based initialization.

To further validate this observation, we grow two neurons in the last layer of the CIFAR10 network. As previously shown in the last row of Fig. 2, the last layer of the network is already able to respond correctly to the inputs. Additionally, the results, shown in Fig. 8b, indicate that the new neurons respond correctly to inputs from different classes. These findings reinforce our conclusion that the effectiveness of newly added neurons depends heavily on the representational quality of the existing neurons.

### D.2.2 DEGENERATION

In Section 4.4, we observed that applying degeneration to the first layer of the CIFAR10 network leads to a substantial drop in overall accuracy. This is because the first layer of the CIFAR10 model contains very few inactive neurons, as visualized in Fig. 2. However, deeper layers—such as layers 3 and 4 in the CIFAR-10 network and layers 5 and 6 in the CIFAR-100 network—exhibit a large number of inactive neurons (i.e., neurons producing near-zero or zero responses). For example,

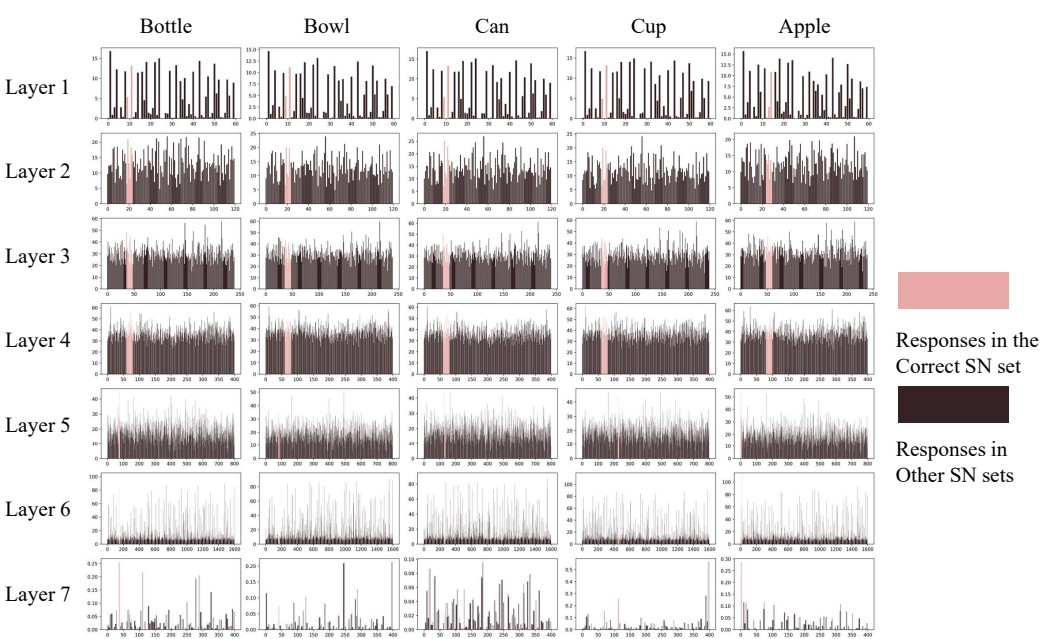

Figure 3: Neurons' responses across layers on CIFAR100.

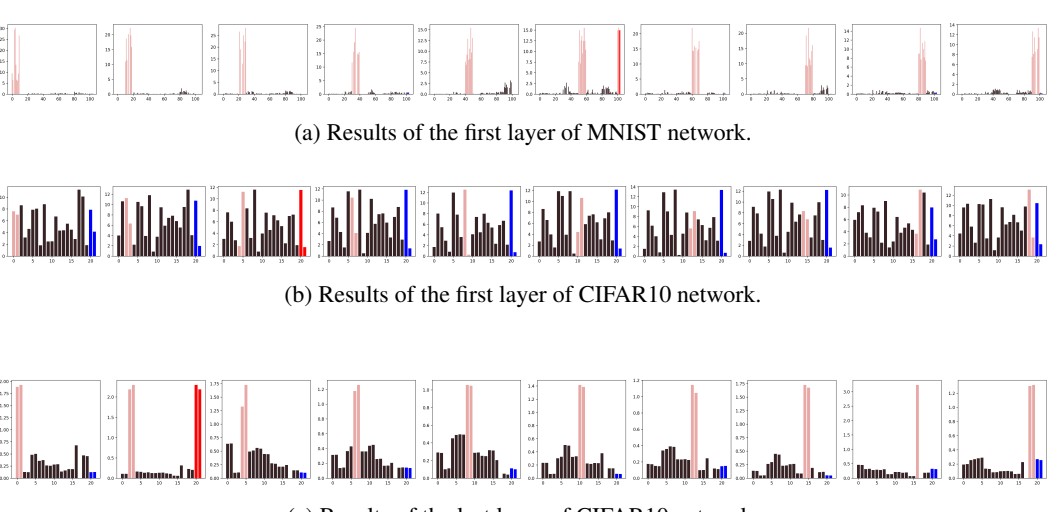

(a) Results of the first layer of MNIST network.

(b) Results of the first layer of CIFAR10 network.

(c) Results of the last layer of CIFAR10 network.

Figure 4: Neurons' responses for neuron growth. In each subfigure, the x axis represents the neuron indices and the bars indicate the corresponding responses.

Fig. 5 provides magnified views of the activations in the last convolutional layers of both networks, where several neurons show minimal or no responses.

Based on this observation, we further examine degeneration on layers that contain more inactive neurons. Specifically, we apply the degeneration algorithm to the last convolutional layers of both the CIFAR10 and CIFAR100 networks, removing neurons one by one. Both networks are well

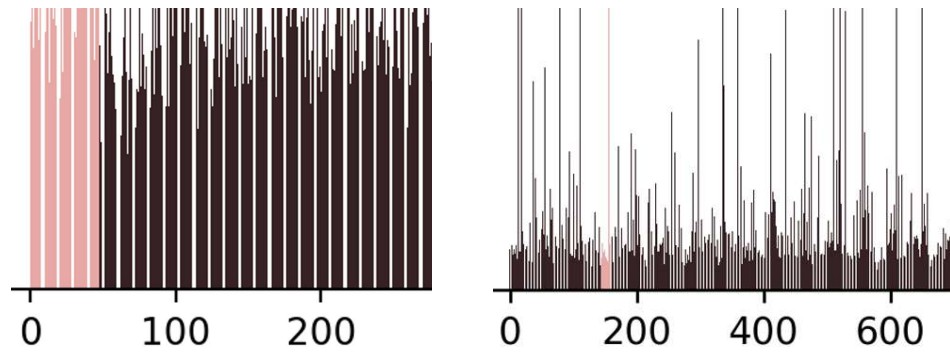

Figure 5: Magnified neuron response patterns. The left visualization shows the last convolutional layer in CIFAR-10, and the right shows the corresponding layer in CIFAR-100.

trained under the default settings. The results are presented in Fig. 6. The dashed lines indicate the test accuracy before any degeneration, while the solid lines depict the accuracy as neurons are progressively removed.

As shown in the figure, the test accuracy remains largely unaffected when only a small number of neurons are degenerated. However, as more neurons are removed, performance gradually deteriorates, indicating that useful neurons are eventually being eliminated. These findings further highlight the effectiveness of the degeneration algorithm: layers that contain inactive or weakly responsive neurons can be pruned without significantly harming performance.

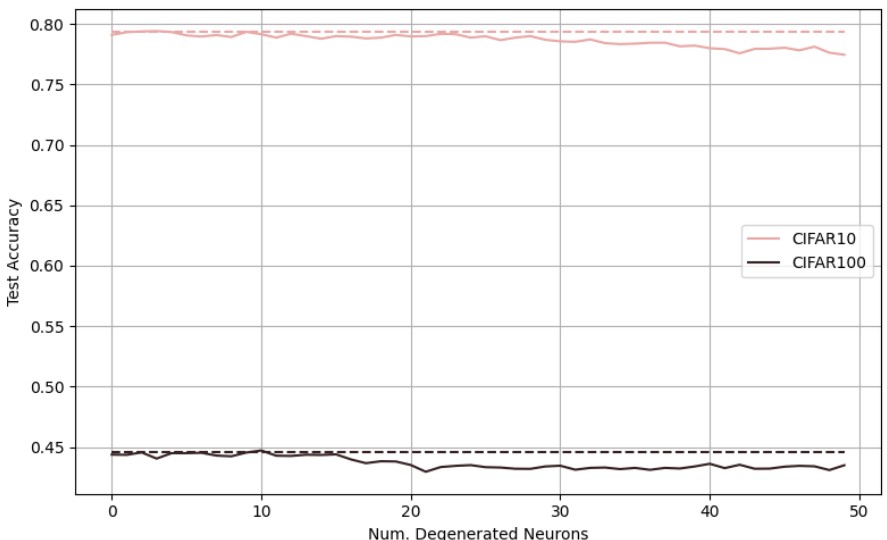

Figure 6: Degeneration performance.

## D.3 SENSITIVITY ANALYSIS

We also conduct sensitivity analyses for several key parameters.

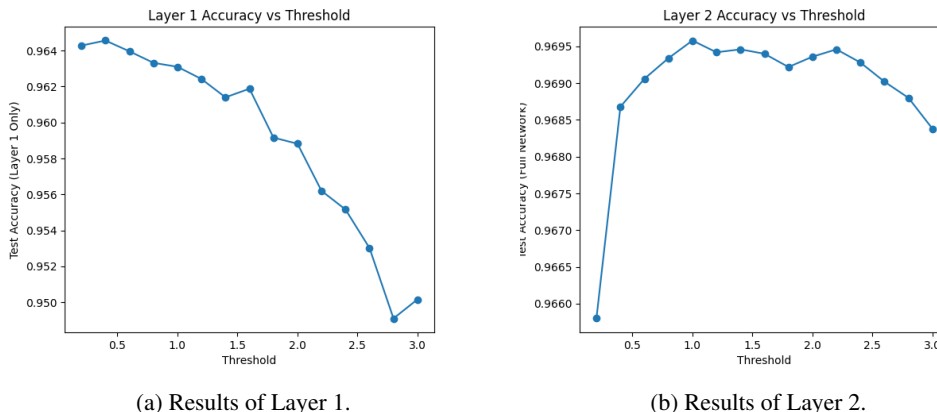

(a) Results of Layer 1.                    (b) Results of Layer 2.

Figure 7: PvN threshold vs. Test Accuracy.

### D.3.1 PvN THRESHOLD

We use the fully connected network on MNIST to investigate how the PvN threshold affects performance. We vary the threshold and report the test accuracy. The first layer is straightforward to obtain the results; for the second layer, however, we fix the threshold of the first layer at 2 and then vary only the second-layer threshold. The results are shown in Fig. 7. Although the threshold does influence performance and the degree of influence differs across layers, the overall fluctuation remains small (less than 1%). Interestingly, the behaviors of the two layers are almost opposite: accuracy decreases in the first layer but increases in the second layer as the threshold grows. This may be because the first-layer neurons are not strong enough to respond under a high threshold, whereas the second-layer neurons are trained to be sufficiently strong. A similar phenomenon can be observed in Fig. 2a, where the neurons in the second layer exhibit stronger responses than those in the first.

### D.3.2 DEGENERATION THRESHOLD

We analyze the effect of the degeneration threshold using the last convolutional layers of the CIFAR10 and CIFAR100 networks. As observed earlier, only a few strong neurons in the correct SN set exhibit larger responses than the others in these layers, and the number of neurons is sufficiently large for meaningful analysis. We define the degeneration threshold $th$ as a percentage: for each class $k$, if the class-wise response of a neuron is small or less than $th \cdot g_{k,\max}$, where $g_{k,\max}$ is the maximum response within SN set $k$, then that neuron is degenerated. The results are presented in Fig. 8.

As illustrated, the behaviors are again almost opposite. In CIFAR10, only a small number of neurons are deleted initially, so the performance is barely affected (consistent with Fig. 6). However, as the threshold increases, more important neurons get degenerated, leading to a performance drop. In contrast, in CIFAR100, the performance increases as the threshold grows and eventually converges, though a performance gap remains compared to the non-degenerated case. This difference arises because, in the last convolutional layer, although both networks rely on a few strong neurons, the remaining weaker neurons behave differently, as shown in the last row of Figs. 2b and 3. In CIFAR10, the responses of the other neurons are relatively close to those of the strongest neurons, whereas in CIFAR100 the differences are much larger thus the less strong neurons may have negative impact on the performance. Consequently, even at $th = 0.05$, 1208 out of 1600 neurons are degenerated in CIFAR100, while none are removed in CIFAR10 at the same threshold. This phenomenon may indicate that many neurons in overparameterized networks are redundant, further highlighting the potential of our degeneration method.

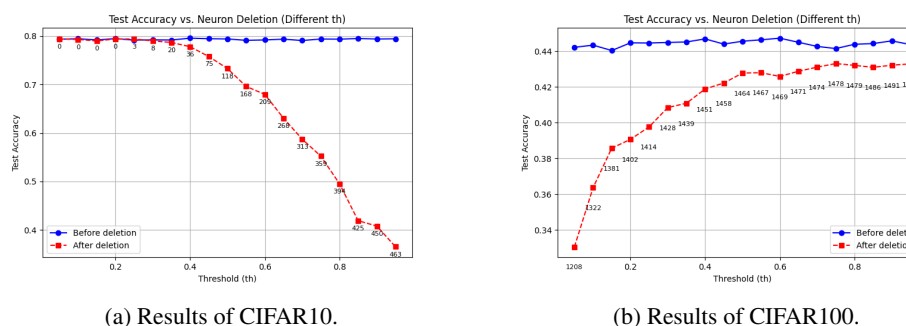

(a) Results of CIFAR10.          (b) Results of CIFAR100.

Figure 8: Degeneration threshold vs. Test Accuracy. The number under the red line means the number of degenerated neurons

### D.4 TRAINING DYNAMICS

The training dynamics, including both loss and accuracy, are illustrated in Fig. 9. All results are based on a single run. As shown in the figure, training generally becomes more stable as the network depth increases.

For MNIST, only the PvN loss exhibits a steadily improving accuracy in the first layer and achieves the best performance across both layers. In contrast, the other loss functions behave unstably and perform poorly on this dataset. For both CIFAR-10 and CIFAR-100, however, the PvN loss becomes the worst among PvN, CE, and CWC in the convolutional layers. Notably, the first 4 layers of the CIFAR-100 architectures are trained on the superclass, thus presenting a low accuracy in that layers. Although the CWC loss performs well in the convolutional stages, its performance drops sharply in the final linear layer after several epochs, indicating instability when applied to linear layers. Additionally, the sparse CE loss consistently exhibits unstable training dynamics and inferior accuracy. Importantly, the Combo training strategy always achieves the best performance in the last layer accuracy.

These observations further support our earlier conclusion that the PvN loss may be unsuitable for convolutional layers, whereas the CWC loss may not be suitable for linear layers.

### D.5 EXTRA RESULTS OF NEURON GROWTH AND DEGENERATION

To facilitate a more comprehensive interpretation of growth and degeneration, we present additional experimental results. Each experiment was conducted at least four times, and key outcomes are emphasized in bold for enhanced clarity.

Table 5: More results of degeneration on MNIST.

| Run | Num. of Del. | Acc. | Acc. 1 | Acc. 2 | Acc. 3 | Acc. 4 | Acc. 5 |
|---|---|---|---|---|---|---|---|
| 1 | 0 | 75.95 | 70.00 | 94.27 | 86.34 | 74.75 | 91.85 |
| 1 | 1 | 78.19 | **95.31** | 94.27 | 85.95 | 74.55 | 91.85 |
| 1 | 2 | 76.68 | **98.88** | 94.27 | 82.66 | 70.79 | 78.19 |

| Run | Num. of Del. | Acc. | Acc. 6 | Acc. 7 | Acc. 8 | Acc. 9 | Acc. 10 |
|---|---|---|---|---|---|---|---|
| 1 | 0 | 75.95 | 86.88 | 92.59 | 11.87 | 74.74 | 77.21 |
| 1 | 1 | 78.19 | 85.99 | 92.07 | 11.48 | 74.64 | 77.21 |
| 1 | 2 | 76.68 | 80.04 | 89.14 | 10.41 | 73.20 | 76.71 |

| Run | Num. of Del. | Acc. | Acc. 1 | Acc. 2 | Acc. 3 | Acc. 4 | Acc. 5 |
|---|---|---|---|---|---|---|---|
| 2 | 0 | 80.66 | 96.73 | 92.16 | 75.19 | 48.71 | 90.43 |
| 2 | 1 | 79.94 | 96.73 | **96.92** | 72.29 | 47.62 | 90.43 |
| 2 | 2 | 80.23 | 96.22 | 96.56 | **87.40** | 41.19 | 89.92 |

| Run | Num. of Del. | Acc. | Acc. 6 | Acc. 7 | Acc. 8 | Acc. 9 | Acc. 10 |
|---|---|---|---|---|---|---|---|
| 2 | 0 | 80.66 | 82.62 | 91.65 | 64.88 | 88.09 | 76.91 |
| 2 | 1 | 79.94 | 80.94 | 91.02 | 63.91 | 83.16 | 76.31 |
| 2 | 2 | 80.23 | 80.38 | 89.56 | 62.84 | 81.42 | 76.21 |

| Run | Num. of Del. | Acc. | Acc. 1 | Acc. 2 | Acc. 3 | Acc. 4 | Acc. 5 |
|---|---|---|---|---|---|---|---|
| 3 | 0 | 84.04 | 91.84 | 89.52 | 86.43 | 90.20 | 89.82 |
| 3 | 1 | 83.93 | **98.06** | 89.52 | 85.56 | 89.50 | 89.51 |
| 3 | 2 | 83.69 | 98.06 | **95.86** | 84.21 | 88.61 | 89.51 |

| Run | Num. of Del. | Acc. | Acc. 6 | Acc. 7 | Acc. 8 | Acc. 9 | Acc. 10 |
|---|---|---|---|---|---|---|---|
| 3 | 0 | 84.04 | 80.49 | 93.01 | 78.50 | 73.92 | 66.11 |
| 3 | 1 | 83.93 | 78.36 | 91.23 | 77.63 | 73.41 | 65.81 |
| 3 | 2 | 83.69 | 76.68 | 90.40 | 77.63 | 73.41 | 65.81 |

| Run | Num. of Del. | Acc. | Acc. 1 | Acc. 2 | Acc. 3 | Acc. 4 | Acc. 5 |
|---|---|---|---|---|---|---|---|
| 4 | 0 | 79.20 | 38.88 | 90.48 | 78.39 | 92.48 | 92.97 |
| 4 | 1 | 83.63 | **84.69** | 90.48 | 78.10 | 92.48 | 92.97 |
| 4 | 2 | 84.19 | **97.96** | 90.48 | 76.45 | 91.88 | 92.97 |

| Run | Num. of Del. | Acc. | Acc. 6 | Acc. 7 | Acc. 8 | Acc. 9 | Acc. 10 |
|---|---|---|---|---|---|---|---|
| 4 | 0 | 79.20 | 62.22 | 91.86 | 91.73 | 77.72 | 71.46 |
| 4 | 1 | 83.63 | 62.00 | 91.75 | 91.73 | 77.72 | 71.46 |
| 4 | 2 | 84.19 | 59.53 | 89.67 | 91.73 | 77.00 | 71.26 |

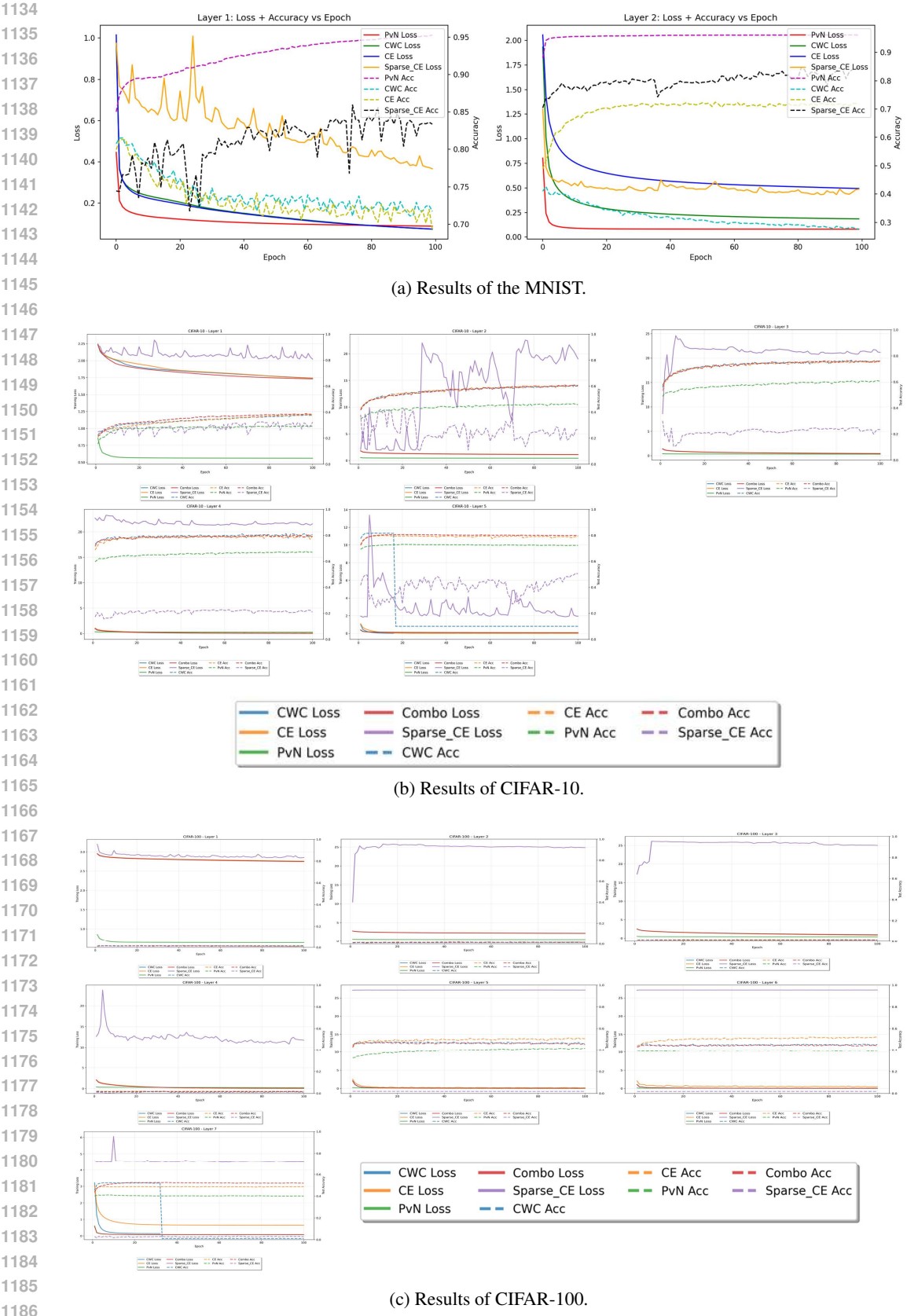

(a) Results of the MNIST.

(b) Results of CIFAR-10.

(c) Results of CIFAR-100.

Figure 9: Training dynamics for loss value and test accuracy.

Table 6: More results of growth on MNIST.

| Run | Net | Acc. | Acc. 1 | Acc. 2 | Acc. 3 | Acc. 4 | Acc. 5 |
|---|---|---|---|---|---|---|---|
| 1 | $net$ | 82.13 | 96.33 | 95.07 | 88.76 | 68.81 | 90.94 |
| 1 | $net_{clone}$ | 83.15 | 96.12 | 94.98 | 88.76 | 62.97 | 90.94 |
| 1 | $net_{train}$ | 83.16 | 96.12 | 94.98 | 88.76 | 61.78 | 90.94 |

| Run | Net | Acc. | Acc. 6 | Acc. 7 | Acc. 8 | Acc. 9 | Acc. 10 |
|---|---|---|---|---|---|---|---|
| 1 | $net$ | 82.13 | 52.91 | 91.13 | 79.86 | 86.04 | 67.59 |
| 1 | $net_{clone}$ | 83.15 | **73.77** | 90.61 | 79.86 | 84.50 | 67.39 |
| 1 | $net_{train}$ | 83.16 | **75.56** | 90.61 | 79.86 | 84.29 | 67.29 |

| Run | Net | Acc. | Acc. 1 | Acc. 2 | Acc. 3 | Acc. 4 | Acc. 5 |
|---|---|---|---|---|---|---|---|
| 2 | $net$ | 84.67 | 94.08 | 88.99 | 85.85 | 90.59 | 81.87 |
| 2 | $net_{clone}$ | 85.13 | 93.57 | 88.81 | 85.85 | 90.30 | 81.87 |
| 2 | $net_{train}$ | 85.14 | 93.57 | 88.81 | 85.85 | 90.10 | 81.87 |

| Run | Net | Acc. | Acc. 6 | Acc. 7 | Acc. 8 | Acc. 9 | Acc. 10 |
|---|---|---|---|---|---|---|---|
| 2 | $net$ | 84.67 | 59.19 | 93.84 | 85.89 | 72.28 | 90.78 |
| 2 | $net_{clone}$ | 85.13 | **66.48** | 93.53 | 85.89 | 71.66 | 90.78 |
| 2 | $net_{train}$ | 85.14 | **67.04** | 93.42 | 85.89 | 71.56 | 90.78 |

| Run | Net | Acc. | Acc. 1 | Acc. 2 | Acc. 3 | Acc. 4 | Acc. 5 |
|---|---|---|---|---|---|---|---|
| 3 | $net$ | 81.54 | 97.14 | 85.29 | 75.39 | 84.06 | 79.74 |
| 3 | $net_{clone}$ | 83.43 | 97.14 | 85.29 | 75.39 | 83.17 | 79.74 |
| 3 | $net_{train}$ | 83.64 | 97.14 | 85.29 | 75.39 | 83.17 | 79.74 |

| Run | Net | Acc. | Acc. 6 | Acc. 7 | Acc. 8 | Acc. 9 | Acc. 10 |
|---|---|---|---|---|---|---|---|
| 3 | $net$ | 81.54 | 30.04 | 94.89 | 84.53 | 91.48 | 87.91 |
| 3 | $net_{clone}$ | 83.43 | **53.03** | 94.68 | 84.53 | 90.97 | 87.91 |
| 3 | $net_{train}$ | 83.64 | **55.49** | 94.68 | 84.53 | 90.86 | 87.91 |

| Run | Net | Acc. | Acc. 1 | Acc. 2 | Acc. 3 | Acc. 4 | Acc. 5 |
|---|---|---|---|---|---|---|---|
| 4 | $net$ | 77.89 | 97.65 | 95.24 | 78.49 | 72.48 | 89.71 |
| 4 | $net_{clone}$ | 81.47 | 97.55 | 95.24 | 78.49 | 71.58 | 89.71 |
| 4 | $net_{train}$ | 81.81 | 97.55 | 95.24 | 78.49 | 71.49 | 89.71 |

| Run | Net | Acc. | Acc. 6 | Acc. 7 | Acc. 8 | Acc. 9 | Acc. 10 |
|---|---|---|---|---|---|---|---|
| 4 | $net$ | 77.89 | 5.94 | 94.89 | 90.27 | 86.24 | 59.27 |
| 4 | $net_{clone}$ | 81.47 | **47.65** | 94.78 | 90.27 | 85.93 | 59.27 |
| 4 | $net_{train}$ | 81.81 | **51.57** | 94.78 | 90.27 | 85.93 | 59.27 |

Table 7: More results of degeneration on CIFAR-10.

| Run | Num. of Del. | Acc. | Acc. 1 | Acc. 2 | Acc. 3 | Acc. 4 | Acc. 5 |
|-----|--------------|-------|--------|--------|--------|--------|--------|
| 1 | 0 | 36.69 | 40.99 | 30.10 | 15.80 | 27.00 | 35.60 |
| 1 | 1 | 18.13 | 32.40 | 7.10 | 1.20 | 0.60 | 1.20 |
| 1 | 2 | 20.05 | 32.30 | 7.00 | 1.50 | 0.60 | **27.40** |

| Run | Num. of Del. | Acc. | Acc. 6 | Acc. 7 | Acc. 8 | Acc. 9 | Acc. 10 |
|-----|--------------|-------|--------|--------|--------|--------|---------|
| 1 | 0 | 36.69 | 35.30 | 58.90 | 32.18 | 54.80 | 36.60 |
| 1 | 1 | 18.13 | 35.30 | 0.00 | **98.60** | 38.90 | 0.70 |
| 1 | 2 | 20.05 | 0.00 | 0.00 | 92.40 | 38.00 | 0.70 |

| Run | Num. of Del. | Acc. | Acc. 1 | Acc. 2 | Acc. 3 | Acc. 4 | Acc. 5 |
|-----|--------------|-------|--------|--------|--------|--------|--------|
| 2 | 0 | 37.61 | 40.80 | 40.60 | 12.20 | 26.10 | 43.90 |
| 2 | 1 | 23.53 | **92.20** | 7.10 | 5.70 | 15.10 | 30.70 |
| 2 | 2 | 16.21 | 85.80 | 1.50 | **66.00** | 0.01 | 0.00 |

| Run | Num. of Del. | Acc. | Acc. 6 | Acc. 7 | Acc. 8 | Acc. 9 | Acc. 10 |
|-----|--------------|-------|--------|--------|--------|--------|---------|
| 2 | 0 | 37.61 | 32.70 | 52.70 | 31.60 | 56.60 | 38.90 |
| 2 | 1 | 23.53 | 16.80 | 44.70 | 16.40 | 0.00 | 6.60 |
| 2 | 2 | 16.21 | 3.70 | 0.60 | 1.80 | 0.00 | 2.20 |

| Run | Num. of Del. | Acc. | Acc. 1 | Acc. 2 | Acc. 3 | Acc. 4 | Acc. 5 |
|-----|--------------|-------|--------|--------|--------|--------|--------|
| 3 | 0 | 38.46 | 40.20 | 45.30 | 14.40 | 24.70 | 40.80 |
| 3 | 1 | 14.98 | 24.00 | 6.30 | 0.10 | 0.10 | **98.30** |
| 3 | 2 | 17.59 | 6.90 | **76.60** | 0.10 | 0.10 | 89.90 |

| Run | Num. of Del. | Acc. | Acc. 6 | Acc. 7 | Acc. 8 | Acc. 9 | Acc. 10 |
|-----|--------------|-------|--------|--------|--------|--------|---------|
| 3 | 0 | 38.46 | 35.90 | 53.60 | 31.20 | 58.90 | 39.60 |
| 3 | 1 | 14.98 | 0.90 | 0.10 | 0.00 | 17.00 | 3.00 |
| 3 | 2 | 17.59 | 0.80 | 0.10 | 0.00 | 1.30 | 0.10 |

| Run | Num. of Del. | Acc. | Acc. 1 | Acc. 2 | Acc. 3 | Acc. 4 | Acc. 5 |
|-----|--------------|-------|--------|--------|--------|--------|--------|
| 4 | 0 | 37.51 | 36.30 | 47.00 | 18.40 | 26.00 | 36.50 |
| 4 | 1 | 16.92 | 27.70 | 12.10 | **92.00** | 0.00 | 0.20 |
| 4 | 2 | 16.83 | 20.10 | 8.80 | 75.50 | **34.40** | 0.00 |

| Run | Num. of Del. | Acc. | Acc. 6 | Acc. 7 | Acc. 8 | Acc. 9 | Acc. 10 |
|-----|--------------|-------|--------|--------|--------|--------|---------|
| 4 | 0 | 37.51 | 30.10 | 56.40 | 29.10 | 62.20 | 33.10 |
| 4 | 1 | 16.92 | 0.20 | 0.10 | 0.00 | 30.00 | 0.00 |
| 4 | 2 | 16.83 | 0.00 | 0.00 | 0.00 | 26.90 | 2.60 |

Table 8: More results of growth on CIFAR-10.

| Run | Net | Acc. | Acc. 1 | Acc. 2 | Acc. 3 | Acc. 4 | Acc. 5 |
|-----|-----|------|--------|--------|--------|--------|--------|
| 1 | $net$ | 37.46 | 44.30 | 44.20 | 14.80 | 18.60 | 39.00 |
| 1 | $net_{clone}$ | 37.46 | 44.30 | 44.20 | **14.80** | 18.60 | 39.00 |
| 1 | $net_{train}$ | 37.48 | 44.50 | 44.20 | **18.20** | 18.60 | 36.40 |

| Run | Net | Acc. | Acc. 6 | Acc. 7 | Acc. 8 | Acc. 9 | Acc. 10 |
|-----|-----|------|--------|--------|--------|--------|---------|
| 1 | $net$ | 37.46 | 35.30 | 55.60 | 30.30 | 55.80 | 36.70 |
| 1 | $net_{clone}$ | 37.46 | 35.30 | 55.60 | 30.30 | 55.80 | 36.70 |
| 1 | $net_{train}$ | 37.48 | 35.40 | 54.90 | 30.30 | 55.80 | 36.50 |

| Run | Net | Acc. | Acc. 1 | Acc. 2 | Acc. 3 | Acc. 4 | Acc. 5 |
|-----|-----|------|--------|--------|--------|--------|--------|
| 2 | $net$ | 37.33 | 37.30 | 44.80 | 14.30 | 26.50 | 37.70 |
| 2 | $net_{clone}$ | 37.33 | 37.30 | 44.80 | **14.30** | 26.50 | 37.70 |
| 2 | $net_{train}$ | 37.41 | 37.50 | 44.80 | **15.60** | 26.60 | 36.60 |

| Run | Net | Acc. | Acc. 6 | Acc. 7 | Acc. 8 | Acc. 9 | Acc. 10 |
|-----|-----|------|--------|--------|--------|--------|---------|
| 2 | $net$ | 37.33 | 33.30 | 54.90 | 28.50 | 61.80 | 34.20 |
| 2 | $net_{clone}$ | 37.33 | 33.30 | 54.90 | 28.50 | 61.80 | 34.20 |
| 2 | $net_{train}$ | 37.41 | 33.50 | 54.90 | 28.50 | 61.90 | 34.20 |

| Run | Net | Acc. | Acc. 1 | Acc. 2 | Acc. 3 | Acc. 4 | Acc. 5 |
|-----|-----|------|--------|--------|--------|--------|--------|
| 3 | $net$ | 36.69 | 38.90 | 42.30 | 12.00 | 19.10 | 40.70 |
| 3 | $net_{clone}$ | 36.69 | 38.90 | 42.30 | **12.00** | 19.10 | 40.70 |
| 3 | $net_{train}$ | 36.67 | 39.10 | 42.30 | **18.80** | 19.20 | 34.30 |

| Run | Net | Acc. | Acc. 6 | Acc. 7 | Acc. 8 | Acc. 9 | Acc. 10 |
|-----|-----|------|--------|--------|--------|--------|---------|
| 3 | $net$ | 36.69 | 37.30 | 56.30 | 32.70 | 53.00 | 34.60 |
| 3 | $net_{clone}$ | 36.69 | 37.30 | 56.30 | 32.70 | 53.00 | 34.60 |
| 3 | $net_{train}$ | 36.67 | 37.10 | 55.80 | 32.50 | 53.00 | 34.60 |

| Run | Net | Acc. | Acc. 1 | Acc. 2 | Acc. 3 | Acc. 4 | Acc. 5 |
|-----|-----|------|--------|--------|--------|--------|--------|
| 4 | $net$ | 37.95 | 40.50 | 48.30 | 11.90 | 22.60 | 42.30 |
| 4 | $net_{clone}$ | 37.95 | 40.50 | 48.30 | **11.90** | 22.60 | 42.30 |
| 4 | $net_{train}$ | 37.96 | 40.90 | 48.30 | **14.70** | 22.80 | 39.00 |

| Run | Net | Acc. | Acc. 6 | Acc. 7 | Acc. 8 | Acc. 9 | Acc. 10 |
|-----|-----|------|--------|--------|--------|--------|---------|
| 4 | $net$ | 37.95 | 38.50 | 54.40 | 29.40 | 55.80 | 35.80 |
| 4 | $net_{clone}$ | 37.95 | 38.50 | 54.40 | 29.40 | 55.80 | 35.80 |
| 4 | $net_{train}$ | 37.96 | 38.50 | 54.30 | 29.40 | 55.90 | 35.80 |

| Run | Net | Acc. | Acc. 1 | Acc. 2 | Acc. 3 | Acc. 4 | Acc. 5 |
|-----|-----|------|--------|--------|--------|--------|--------|
| 5 | $net$ | 36.85 | 43.70 | 40.10 | 14.00 | 19.70 | 43.70 |
| 5 | $net_{clone}$ | 36.85 | 43.70 | 40.10 | **14.00** | 19.70 | 43.70 |
| 5 | $net_{train}$ | 36.88 | 44.20 | 40.10 | **19.10** | 19.60 | 38.60 |

| Run | Net | Acc. | Acc. 6 | Acc. 7 | Acc. 8 | Acc. 9 | Acc. 10 |
|-----|-----|------|--------|--------|--------|--------|---------|
| 5 | $net$ | 36.85 | 38.30 | 52.90 | 28.50 | 46.40 | 41.20 |
| 5 | $net_{clone}$ | 36.85 | 38.30 | 52.90 | 28.50 | 46.40 | 41.20 |
| 5 | $net_{train}$ | 36.88 | 38.20 | 52.80 | 28.50 | 46.50 | 41.20 |