# OpenReview forum: "Forward-Forward Learning with Dynamic Architecture Adaptation for Classification"
_ICLR.cc/2026/Conference — Submitted to ICLR 2026_

### Official Review · Reviewer_GDeD · 2025-10-25

**Soundness:** 2
**Presentation:** 3
**Contribution:** 2
**Rating:** 2
**Confidence:** 3

**Summary:**

This paper proposes a Forward-Forward (FF)-based learning framework that introduces support neuron (SN) sets to partition neurons by class, thereby eliminating the need for negative samples. Furthermore, the authors design biologically inspired neuron growth and degeneration mechanisms to dynamically adjust network capacity. \

**Strengths:**

- The paper explores an interesting direction by combining FF learning with dynamic neural structural adaptation.

- The biologically inspired neuron growth and degeneration mechanisms are interesting for dynamically adjusting network capacity.

**Weaknesses:**

**Ambiguity in the Combo Training Process (Table 1)**

It is unclear whether the training process using CWC for convolutional layers and PvN for linear layers is conducted sequentially (train convolutional layers first, then linear layers) or simultaneously (both optimized in the same iteration). Moreover, since the proposed method avoids standard backpropagation, it is essential to clarify how parameter updates are coordinated between these two objectives.

**Inconsistencies in Reported Results (Table 2)**

The reported accuracies for FF and CaFo on MNIST appear inconsistent with values presented in the respective original papers. Additionally, FF is only examined on MNIST is not precise. Both FF and CaFo have published convolutional and non-convolutional variants, and original results for CIFAR-10/100 are available.

**Missing Training Dynamics and Visualization **

The paper lacks visual or statistical evidence regarding the training dynamics and representation evolution. For instance, it would significantly strengthen the work to include:

- Plots of training and validation accuracy versus epochs to demonstrate convergence behavior and stability.
- Visualization of response distribution across SN sets over time to support claims about interpretability and neuron specialization.
- Optional activation heatmaps showing how neurons evolve before and after growth/degeneration.


**Theoretical Rationality of Explicit Class-wise SN Assignment**

The core assumption that neurons can be explicitly and statically assigned to specific classes deserves deeper theoretical justification. In realistic scenarios, classes often share overlapping features; thus, forcing neurons to be class-exclusive may restrict feature sharing and harm generalization.

**Minor Comments**
1. The description of  “Combo” training (CWC + PvN) could include pseudocode for clarity.
2. Consider adding figures illustrating neuron growth/degeneration workflows.
3. The transition from the critical analysis to the overall assessment is not sufficiently smooth, which makes the logical flow of evaluation appear somewhat abrupt and may weaken the coherence and persuasiveness of the review.

**Questions:**

See above.

---

> ### Author Response · Authors · 2025-11-22
>
> Thank you very much for your valuable comments. Below we provide point-by-point responses.
>
> **$\textbf{1. Ambiguity in Combo Training}$**
>
> Due to space limitations, we provided the training pseudo-code in the Appendix A. In our proposed method, the model is trained layer-by-layer, and each layer is optimized sequentially. This design naturally allows us to adopt different loss functions for different layers.
>
> **$\textbf{2. Inconsistencies in Reported Results}$**
>
> a. The performance differences between FF and CaFo arise from adjustments in the network architecture and training protocol (e.g., data preprocessing and augmentation). We experimented with two architectures for CaFo: one from the original paper and one aligned with our proposed design, and we retained the version that achieved the best performance.
>
> To ensure a fair comparison across methods, we intentionally kept the training pipeline as simple as possible—for example, we used no data augmentation beyond basic normalization—so that the results are not biased by hyperparameter tuning.
>
> b. We are currently conducting additional experiments on variants of FF, including (but not limited to) convolutional FF models. These results will be included in the revised submission.
>
> c. Experiments on training dynamics and visualization are also underway. We will update all corresponding results in the revised version.
>
> **$\textbf{3. Theoretical Rationale for Explicit Class-wise SN Assignment}$**
>
> We agree with the reviewer that having overlapping SN sets between classes is theoretically reasonable. However, manually designing such overlaps is not feasible.
>
> First, as the number of classes increases, crafting appropriate overlaps between every pair of classes becomes intractable.
> Second, enforcing uniform overlap across class pairs is inappropriate; for example, the overlap between “white cat” and “black cat” should naturally be larger than that between “white cat” and “white car.”
>
> Therefore, an automated mechanism would be necessary to capture such structure.
>
> We attempted the following reorganization strategy:
> ```
> Reorganization (per layer)
> For each class k among the K classes:
> • Input class-k data to the layer
> • Record each neuron’s average response
> • Record the average response of the k-th SN set
> • If a neuron not in SN set k shows an average response higher than the SN-set average, add it to SN set k
> ```
> Unfortunately, this automatic overlap mechanism did not yield improvements and in many cases degraded performance, regardless of whether it was applied after training or integrated into the training loop.
> We believe there may exist more effective strategies for designing overlaps, and we have added this as a future research direction in Section 5 (Conclusion and Perspectives).
>
> **$\textbf{4. Minor Comments}$**
>
> We appreciate the your detailed suggestions and will carefully address all minor comments in the revised manuscript.

---

> ### Author Response · Authors · 2025-11-25
>
> Before we begin, we would like to address an inconsistency in the reported CaFo results. We previously overlooked the fact that CaFo was trained for significantly more epochs in the original paper—5000 epochs for MNIST and CIFAR-10, and 1000 epochs for CIFAR-100—whereas our replication used only 100 epochs for all datasets. This discrepancy likely accounts for the difference in results. We apologize for this oversight.
>
> According to your valuable comments, we have conducted additional experiments and revised the manuscript accordingly. First, we visualized the training dynamics, including loss values and test accuracies across training epochs. We then visualized neuron responses across layers by feeding samples from different classes separately and plotting per-neuron activations (Appendix D.1). For MNIST and CIFAR-10, we analyzed class-wise neuron behaviors; for CIFAR-100, we investigated cross-class behaviors among several highly related classes as well as a less related one. We believe these results help enhance the explainability of our method.
>
> Moreover, we illustrated how neurons evolve after growth. For degeneration, we observed the presence of inactive neurons in some deep layers and evaluated the overall accuracy while removing these neurons one by one. This analysis demonstrates the practical impact of degeneration and the robustness of the model to moderate neuron removal. Also, we have done several works according to other reviewers’ comments.
> All revisions are highlighted in blue for your convenience, and the color will be removed in the final submission. The main revisions are summarized as follows:
> 1. We revised parts of the main manuscript to give proper credit to the references and to further clarify our contributions, aiming to avoid potential confusion. Besides, we slightly revised the training pseudo code for clarity.
> 2. We added additional experiments to visualize latent activations across all layers (Appendix D.1). These include class-wise response analysis on MNIST and CIFAR-10, as well as cross-class behavior analysis among several highly related classes on CIFAR-100. We believe these results enhance the explainability of neuron behaviors.
> 3. We added further experiments on neuron growth and degeneration (Appendix D.2). First, we visualized neuron responses after growth to analyze the behavior of new neurons. Second, we observed that deep layers contain more inactive neurons than the first layer, particularly in the CIFAR-10 experiments. We further illustrate overall accuracy versus degenerated neurons to demonstrate the practical impact of degeneration.
> 4. We added experiments examining the training dynamics, including loss values and test accuracy (Appendix D.3). These results not only depict the training behavior but also support the previous conclusion that the PvN loss is more suitable for linear layers, while the CWC loss is more suitable for convolutional layers. Due to computational constraints, these results are based on a single run; we plan to replace them with averages over multiple runs in a future version.
> 5. We added experiments for a new loss, CE with Sparsemax function, to further distinguish it from the CWC loss (Section 4.2). In addition, we corrected a mis-recording of the CaFo method and revised the MNIST architecture accordingly.
> 6. We added experiments on convolutional version of FF (in Section 4.3).
>
> The above results are not exhaustive. Ongoing work includes:
>
> 1. Accuracy evaluation of other comparison algorithms, including non-BP algorithms and FF variants.
>
> 2. Sensitivity analysis for certain hyperparameters.
>
> Furthermore, we plan to include additional details in the manuscript, such as pseudocode for the growth and degeneration procedures.
>
> We submit the current revised version, which does not yet include all results, with the goal of demonstrating that the newly added results significantly improve the quality of the paper. As there is still time before the rebuttal deadline, we hope this allows reviewers sufficient time for evaluation. We apologize for the delay in the revised manuscript and regret any potential inconvenience this may cause to the reviewers.

---

> > ### Comment · Reviewer_GDeD · 2025-11-26
> >
> > Thank you for your clarification. First of all, there still some following related questions regarding your reply:
> >
> > 1.	Ambiguity in Combo Training.
> > We have reviewed Appendix A and confirm that your proposed method indeed adopts a layer-wise training strategy, where the model is trained sequentially layer by layer. This design choice naturally facilitates the use of distinct loss functions tailored to individual layers, which is an interesting aspect of your approach.
> >
> > 2.	Inconsistencies in Reported Results.
> > While the authors’ effort to align baseline architectures with their proposed method for fair comparison is understandable, several concerns remain regarding the selection and reporting of baselines:
> >
> > a. Selective Reporting of CaFo Variants:
> > Notably, on CIFAR-10/100, the reported results are consistent with applying the authors’ unified architecture and training protocol to the original CaFo model. This demonstrates that integrating the original CaFo implementation into their framework is indeed feasible—even on more complex datasets. This raises a critical inconsistency: if the original CaFo can be successfully trained under the authors’ simplified protocol on CIFAR-10/100 and yields performance aligned with the original paper, then it should also be straightforward to evaluate it under the same protocol on MNIST. Yet, only a weaker, architecture-modified variant is reported for MNIST. If the original CaFo—when trained with the authors’ protocol—shows a significant drop in accuracy on MNIST compared to its originally published result, this discrepancy must be explicitly quantified and explained. Otherwise, the selective omission risks undermining the perceived fairness of the comparison.
> >
> > b.Missing Reproduction of FF on CIFAR under Comparable Settings:
> > The paper does not report results for the FF baseline on CIFAR-10/100 under the same non-convolutional or simplifier forward-only learning studies. Omitting these results creates an incomplete baseline picture and weakens the empirical evaluation.
> >
> > 3. Theoretical Rationale for Explicit Class-wise SN Assignment.
> > We believe that the proposed architecture—where neurons’ responses are governed by overlapping SN sets may introduce new challenges. Specifically, dynamically reassigning neuron memberships in deeper layers can lead to internal representation inconsistencies. Since the parameters of earlier layers are fixed once trained, frequent reassignment of neuron roles in subsequent layers may disrupt the semantic alignment and feature coherence between consecutive layers. This instability could potentially impair the model’s overall representational capacity and degrade its generalization performance. While the motivation for allowing overlaps among SN sets is well-founded, we recommend that the authors conduct a more thorough analysis of how neuron reassignment affects model stability and generalization. Alternatively, incorporating a cross-layer coordination mechanism could help mitigate the drawbacks inherent in purely layer-wise sequential optimization.

---

> ### Author Response · Authors · 2025-11-28
>
> Thank you very much for your reply. Before start, We have updated the fully revised manuscript according to the reviewers’ valuable comments. Specifically, compared to the previous version, we have added new comparison results for FF variants (e.g., DeeperForward) as well as non-BP and biologically implausible methods (DRTP and BWBPF). Moreover, we have included sensitivity analyses on both the PvN threshold and the degeneration threshold. Overall, the sensitivity analysis shows that although the PvN threshold influences performance, the resulting fluctuations remain small and layer-dependent. In addition, the results on the degeneration threshold indicate that overparameterized networks contain many weak or redundant neurons, underscoring the potential of our degeneration-based approach.
>
> The followings are the point-to-point responses:
>
> **$\textbf{Selective reporting of CaFo:}$**
>
> Our goal is to evaluate the effectiveness of the proposed training method on both linear and convolutional layers. The MNIST dataset is relatively simple for deep neural networks, and a convolutional neural network can easily achieve over 98\% accuracy. In contrast, a relatively shallow fully connected network is less capable of achieving high performance, which makes it suitable for demonstrating the effectiveness of linear-layer designs. Therefore, we design the experiments as follows:
>
> (1) we evaluate only fully connected neural networks containing only linear layers on the MNIST dataset, and (2) we evaluate only convolutional neural networks on the CIFAR datasets. The original CaFo implementation on MNIST is a convolutional network, so we modify the architecture to compare the effectiveness of linear-layer designs. We revised the corresponding section in the manuscript to further clarify our purpose, together with the new results from additional comparison algorithms. Please find them in our new manuscript.
>
> **$\textbf{Missing reproduction of FF on CIFAR:}$**
>
>  As mentioned above, we aim to assess the effectiveness of our proposed method on convolutional layers for the CIFAR datasets. Thus, we use the convolutional variant of the FF algorithm rather than the original linear version. All methods evaluated on CIFAR use convolutional neural network architectures; no non-convolutional models are used.
>
> **$\textbf{Theoretical rationale for explicit class-wise assignment:}$**
>
> We further investigated the cause behind the failure of the proposed Reorganization method. Specifically, we conducted the following experiment:
> 1. Train a single layer on MNIST for one epoch;
> 2. Apply the Reorganization procedure;
> 3. Collect all reorganized neurons. For each neuron $i$, assume it originally belongs to set $k$ and is reorganized into set $k'$. This neuron is therefore an overlapping neuron between sets $k$ and $k'$. We then feed the data of class $k'$ and compute the percentage of samples for which neuron $i$ produces a stronger response than the class-wise average response of the original set $k'$.
>
> The training accuracy before and after Reorganization is 81.34\% and 75.34\%, respectively, with 11 neurons reorganized. Surprisingly, for every reorganized neuron, fewer than 25\% of the samples in the new class produce a stronger response than the class-wise average response of the original set $k'$. This indicates that these neurons respond correctly to only a small portion of the new class. However, because their correct responses are unusually strong, the class-wise average response becomes high enough to trigger the Reorganization function. Since most samples are not correctly responded to by these neurons, the accuracy decreases.
>
> The current reorganization strategy for overlapping neurons requires further investigation. A potential direction is to integrate neuron allocation and selection into the training phase. Specifically, given a well-defined loss function $L$ and layer $f_l(\cdot \mid \theta)$, we may design an update rule that allows overlapping based on the following optimization problem:
>
> $$
> \min_{\{N_k^l\},\,\theta} \; L(f(x \mid \theta), y)
> $$
> where $N_k^l$ denotes the $k$-th SN set in the $l$-th layer.

---

### Official Review · Reviewer_Hfab · 2025-10-28

**Soundness:** 2
**Presentation:** 3
**Contribution:** 2
**Rating:** 4
**Confidence:** 3

**Summary:**

This paper proposes an extension to the Forward-Forward (FF) learning paradigm for classification. The main innovations are (1) partitioning neurons into class-specific Support Neuron (SN) sets and defining SN-set responses (goodness) as the sum of squared activations for class prediction, and (2) introducing biologically inspired neuron growth and degeneration mechanisms that dynamically adjust network width. The authors evaluate three loss functions, i.e. Positive-vs-Negative (PvN), a channel-wise competitive (CwC) SN-set-wise loss, and Cross-Entropy (CE), and a hybrid training scheme (Combo) that uses CwC for convolutional layers and PvN for the final linear layer. Experiments on MNIST, CIFAR-10, and CIFAR-100 show improvements over other FF-based methods that avoid negative samples (e.g., CFSE and CaFo).

**Strengths:**

Originality: Extending channel/grouping ideas to class-specific Support Neuron (SN) sets and integrates neuron growth/degeneration.

Clarity of core idea: SN-set response (sum of squared activations) and the LLR/OLR prediction criteria are clearly formulated.

Practical relevance: Eliminating negative samples addresses a practical limitation of many FF variants and may simplify implementations for resource-constrained settings.

Empirical promise: The Combo training strategy shows competitive results compared to other FF-based approaches in the paper.

**Weaknesses:**

Motivation: The proposed strategies, e.g. neuron growth and degeneration, seems to be ad hoc, i.e. it lacks a rigorous motivation or toy experiment to validate them.

Baseline fairness: The reported DNN baseline (65.4% on CIFAR-100) appears a little low for comparable architectures. The authors should clarify the DNN architecture, training recipe, and whether data augmentation or other standard techniques were used.

Limited comparison scope: Important recent FF variants and stronger BP baselines (beyond ResNet18 and a single DNN) are not included; this weakens the claim of its superiority.

Insufficient justification for SN response metric and hyperparameters: The square-sum choice needs empirical comparisons (e.g., L1, max, soft-assignment) and sensitivity analysis.

Growth/degeneration analysis: The degradation experiments show large and non-monotonic changes (e.g., removing two neurons sometimes collapses accuracy). Authors should analyze per-neuron contribution and report variance across different random seeds.

Reproducibility: Some low-level training details (exact optimizer schedules, seed handling, and implementation variants for baselines) should be moved into the main text or a reproducibility checklist.

**Questions:**

1. Could the motivation of the some strategies be presented in a more rigorous way ?

2. Please provide the exact DNN baseline architecture and full training recipe for CIFAR-100 (data augmentation, optimizer, learning-rate schedule, number of runs), since the proposed baseline is a little low.

3. Please analyze the neuron removal experiments more thoroughly. For the cases where removing two neurons causes a disproportionate drop, can the authors (a) report per-neuron activation statistics, (b) visualize learned filters or class-specific activations, and (c) show results averaged across multiple random seeds?

4. Have the authors tried alternative SN-set response metrics (e.g., L1 norm of activations, max pooling within channels, cosine similarity on pooled features) ?

5. Please include additional baselines: (i) stronger BP baselines (well-tuned DNN/ResNet variants), and (ii) recent FF variants such as DeeperForward/Distance-Forward if implementations are available.

6. Please provide a sensitivity study for key hyperparameters: thresholds for PvN, selection rules for growth, number of neurons added per growth step, and degeneration thresholds.

7. How about the robustness of the Combo scheme (CwC for conv layers + PvN for linear layers) across architectures and dataset ? An ablation would help.

---

> ### Author Response · Authors · 2025-11-22
>
> Thank you very much for your comments. Below we provide point-to-point response:
>
> **$\textbf{Comment: }$**
>
> Could the motivation of the some strategies be presented in a more rigorous way ?
>
> **$\textbf{Response:}$**
>
> The proposed mechanisms are indeed more of an inspiration drawn from biological observations. Our motivation for growth comes from the well-known fact that the human brain is capable of forming new connections. Since we organize neurons into different sets corresponding to different tasks, it is natural to ask whether, when performance on a given task deteriorates, we can mimic this biological behavior by generating additional connections or neurons to assist learning. Conversely, the motivation for degeneration stems from practical considerations regarding model size: if the network already performs sufficiently well, then it may be possible to reduce complexity by removing neurons—particularly from those SN sets whose associated class-wise accuracy is extremely high.
>
> Although this line of thinking is less rigorous than conventional machine learning formulations, we hope that our method may inspire further exploration into explainable, brain-inspired modeling approaches, such as the work presented in [R1].
> ```
> [R1] Zhao H, Wu H, Yang D, et al. BriLLM: Brain-inspired Large Language Model. arXiv:2503.11299, 2025.
> ```
>
> **$\textbf{Comment:}$**
>
> Please provide the exact DNN baseline architecture and full training recipe for CIFAR-100 (data augmentation, optimizer, learning-rate schedule, number of runs), since the proposed baseline is a little low.
>
> **$\textbf{Response:}$**
>
> Due to space constraints, some experimental details were placed in Appendix A. For your convenience, we summarize them here:
>
> a. Network architecture.
>
> For CIFAR-10 and CIFAR-100, the model consists of a sequence of convolutional blocks. Each block, denoted as Block(in, out, Pool), contains a convolutional layer, a ReLU activation, batch normalization, and an optional max-pooling layer. The convolution uses kernel size 3, stride 1, and padding 1. Max-pooling (when enabled) uses kernel size 2 and stride 2.
>
> For CIFAR-100, the full architecture is:
> Block(3, 60, False) → Block(60, 120, True) → Block(120, 240, False) → Block(240, 400, True) → Block(400, 800, False) → Block(800, 1600, True) → Linear(−1, 400).
>
> b. Data preprocessing.
>
> No data augmentation is used. We only apply:
> transforms.Normalize((0.5071, 0.4867, 0.4408),
>                      (0.2675, 0.2565, 0.2761))
>
> c. Optimization details.
>
> We use Adam with learning rate 0.001 and weight decay 1e-4. The learning rate scheduler has a step size of 10 and gamma = 0.5.
> All experiments are repeated 10 times, and we report the mean accuracy as stated in the main text.
>
> We intentionally keep the training procedure as simple and consistent as possible so that comparisons with our proposed method are fair. Although our method does not outperform highly optimized DNNs, our goal is to evaluate the potential of the proposed brain-inspired framework under identical settings to a comparable DNN.
>
> **$\textbf{Comment: }$**
>
> Please analyze the neuron removal experiments more thoroughly. For the cases where removing two neurons causes a disproportionate drop, can the authors (a) report per-neuron activation statistics, (b) visualize learned filters or class-specific activations, and (c) show results averaged across multiple random seeds?
>
> **$\textbf{Response:}$**
>
> There may have been some misunderstanding. Degeneration experiments involve sequential removal of two neurons: we remove one first, then remove the second.
>
> Regarding averaging across multiple runs: because training outcomes vary across runs, the neurons selected for growth or degeneration may differ. For example, in run 1, the algorithm might degenerate a neuron in SN set 2, whereas in run 2 it degenerates one in SN set 5. In such cases, the SN sets of run 1 and run 2 become structurally different, making direct averaging of class-wise accuracies inappropriate.
>
> We provide multi-run results for both growth and degeneration in Appendix D, which we hope helps evaluate the stability of our method.
>
> Visualization experiments on latent activations are currently in progress and will be included in the revised manuscript.

---

> ### Author Response · Authors · 2025-11-22
>
> **$\textbf{Comment: }$**
>
> Have the authors tried alternative SN-set response metrics (e.g., L1 norm of activations, max pooling within channels, cosine similarity on pooled features) ?
>
> **$\textbf{Response:}$**
>
> Below are the metrics we have tested:
>
> a. L1 norm of activations.
>
> This activation function performed worse than squared activations.
>
> b. SN-set-wise max pooling.
>
> We implemented a top-K-percent pooling mechanism within each SN set. However, this consistently degraded performance.
> For convolutional layers, we also tested flattening the feature maps into one vector and splitting it into SN sets analogous to the linear layer. The pooling mechanism then required resetting low-activation neurons to zero, since direct removal of those neurons altered feature map structures. This design further reduced performance.
>
> c. Randomly selection of neurons.
>
> In our manuscript, we uniformly partition the neurons to obtain the SN sets. We also tested the performance of randomly choose equal number of neurons for each SN set, which perform the same with uniform partition.
>
> **$\textbf{Comment: }$**
>
> Please include additional baselines: (i) stronger BP baselines (well-tuned DNN/ResNet variants), and (ii) recent FF variants such as DeeperForward/Distance-Forward if implementations are available.
>
> **$\textbf{Response:}$**
>
> We agree that a stronger BP baseline would likely outperform the DNN baselines used in our current experiments. Our objective, however, is to demonstrate that the proposed brain-like method has the potential to achieve comparable performance to DNNs under identical training settings, even if some performance gaps remain. Within this scope, we think introducing a stronger BP baseline may not significantly deepen the insight relevant to this specific comparison.
>
> Experiments on other FF-based variants and other non-BP methods are underway, and we will report the results as soon as they are ready.
>
> **$\textbf{Comment: }$**
>
> Please provide a sensitivity study for key hyperparameters: thresholds for PvN, selection rules for growth, number of neurons added per growth step, and degeneration thresholds.
>
> **$\textbf{Response:}$**
>
> We are currently conducting all requested experiments and will submit the updated results soon.
>
> **$\textbf{Comment: }$**
>
> How about the robustness of the Combo scheme (CwC for conv layers + PvN for linear layers) across architectures and dataset ? An ablation would help.
>
> **$\textbf{Response:}$**
>
> We apologize if we misunderstood your concern. Could you please clarify what aspects of the robustness of the combo scheme you would like us to examine (e.g., varying the architecture, comparing with a pure-loss version, or other robustness tests)? We would greatly appreciate more detail so that we can respond thoroughly.

---

> ### Author Response · Authors · 2025-11-25
>
> According to your valuable comments, we have conducted additional experiments and revised the manuscript accordingly. Specifically, we visualized neuron responses across layers by feeding data from different classes separately and plotting the per-neuron activations (Appendix D.1). For MNIST and CIFAR-10, we analyzed class-wise neuron behaviors, while for CIFAR-100, we examined cross-class behaviors among several highly related classes as well as a less related one.
>
> In addition, we performed further investigations on neuron growth and degeneration (Appendix D.2). We observed that certain deep layers in convolutional networks contain inactive neurons, and we demonstrated that removing a moderate number of such neurons does not significantly impact performance. This provides evidence that degeneration can be practical for layers with a large number of parameters. We believe these results help enhance the explainability of our work. Also, we have done several works according to other reviewers’ comments.
> All revisions are highlighted in blue for your convenience, and the color will be removed in the final submission. The main revisions are summarized as follows:
> 1. We revised parts of the main manuscript to give proper credit to the references and to further clarify our contributions, aiming to avoid potential confusion.  .   Besides, we slightly revised the training pseudo code for clarity.
> 2. We added additional experiments to visualize latent activations across all layers (Appendix D.1). These include class-wise response analysis on MNIST and CIFAR-10, as well as cross-class behavior analysis among several highly related classes on CIFAR-100. We believe these results enhance the explainability of neuron behaviors.
> 3. We added further experiments on neuron growth and degeneration (Appendix D.2). First, we visualized neuron responses after growth to analyze the behavior of new neurons. Second, we observed that deep layers contain more inactive neurons than the first layer, particularly in the CIFAR-10 experiments. We further illustrate overall accuracy versus degenerated neurons to demonstrate the practical impact of degeneration.
> 4. We added experiments examining the training dynamics, including loss values and test accuracy (Appendix D.3). These results not only depict the training behavior but also support the previous conclusion that the PvN loss is more suitable for linear layers, while the CWC loss is more suitable for convolutional layers. Due to computational constraints, these results are based on a single run; we plan to replace them with averages over multiple runs in a future version.
> 5. We added experiments for a new loss, CE with Sparsemax function, to further distinguish it from the CWC loss (Section 4.2). In addition, we corrected a mis-recording of the CaFo method and revised the MNIST architecture accordingly.
> 6. We added experiments on convolutional version of FF (in Section 4.3).
>
> The above results are not exhaustive. Ongoing work includes:
>
> 1. Accuracy evaluation of other comparison algorithms, including non-BP algorithms and FF variants.
>
> 2. Sensitivity analysis for certain hyperparameters.
>
> Furthermore, we plan to include additional details in the manuscript, such as pseudocode for the growth and degeneration procedures.
>
> We submit the current revised version, which does not yet include all results, with the goal of demonstrating that the newly added results significantly improve the quality of the paper. As there is still time before the rebuttal deadline, we hope this allows reviewers sufficient time for evaluation. We apologize for the delay in the revised manuscript and regret any potential inconvenience this may cause to the reviewers.

---

> ### Author Response · Authors · 2025-11-28
>
> We have updated the fully revised manuscript according to the reviewers’ valuable comments. Specifically, compared to the previous version, we have added new comparison results for FF variants (e.g., DeeperForward) as well as non-BP and biologically implausible methods (DRTP and BWBPF). Moreover, we have included sensitivity analyses on both the PvN threshold and the degeneration threshold. Overall, the sensitivity analysis shows that although the PvN threshold influences performance, the resulting fluctuations remain small and layer-dependent. In addition, the results on the degeneration threshold indicate that overparameterized networks contain many weak or redundant neurons, underscoring the potential of our degeneration-based approach.
>
> However, as shown in the additional experiments on growth, the newly grown neurons rely heavily on the existing ones, so increasing the number of neurons added at each growth step does not lead to further improvement. Regarding the selection criterion for growth, we tested several thresholds based on the training accuracy, but since the growth procedure does not significantly affect accuracy, all thresholds yield similar results. Therefore, we did not include these results in the manuscript. Nevertheless, we believe that both degeneration and growth merit further investigation in light of the current findings.

---

### Official Review · Reviewer_5spw · 2025-10-31

**Soundness:** 1
**Presentation:** 2
**Contribution:** 1
**Rating:** 0
**Confidence:** 5

**Summary:**

The paper proposes a forward-only, no-negatives training scheme that partitions each layer into per-class Support Neuron (SN) sets and defines “goodness” as sum of squared activations per set. It studies losses on per-class goodness, two prediction rules (LLR, OLR), and introduces neuron growth and neuron degeneration to adapt width. Results are shown on MNIST, CIFAR-10, CIFAR-100, compared to CFSE (Papachristodulou et al, 2024) and CaFo (Zhao et al, 2023) (forward methods without negatives) and to BP baselines.

**Strengths:**

- Width adaptation idea (growth and degeneration) is interesting and could be impactful with deeper evaluation.

**Weaknesses:**

- Conv-layer partitioning into sets is a mechanism already introduced in prior work CFSE (Papachristodulou et al, 2024). The paper acknowledges this but still lists this set partitioning as a core contribution. In my opinion the genuine novelty is the linear-layer extension and width adaptation. In addition, all three losses have already been introduced in prior work with the same name (Papachristodulou et al, 2024). The PvN and CwC as used here are already defined in CFSE (Papachristodulou et al, 2024) for channel groups (CFSE Eqs. 5–6). The manuscript should be explicit about reuse and position its difference. As it stands the mathematical formulation is the same.
- The paper’s CwC equals CE applied to per-class goodness logits (g_l). The CE section repeats the same. This duplication confuses the contribution and makes the loss study harder to interpret.
- Section 3.7 claims training each layer “directly on the entire dataset” and that optimization is “essentially closed-form.” The appendix algorithm trains with random mini-batches using SGD/Adam, i.e., not closed-form nor full-batch.
- Only last-layer edits, small additions/deletions, and sensitivity (including degradations) are shown. Authors note single runs with fluctuating results in some cases.
- Reported gains over CFSE/CaFo are rather incremental (e.g., CIFAR-10 80.5% vs CFSE 77.2%; CIFAR-100 52.0% vs CFSE 48.9%). This is fine, but it does not show a qualitative jump.
- The paper states “architectures are kept consistent” in Table 2 comparisons, but also evaluates CaFo with a revised architecture (Re). This change should be labeled carefully.
- The paper lacks comparison and/or discussion with standard local-learning and FF-inspired methods beyond CFSE/CaFo. In particular FA, DFA, DRTP, SoftHebb. These are canonical non-BP or local-error baselines that target similar motivations (forward compute, cheap feedback, biological plausibility). The paper is missing a comparison with at least one method per family or provide a documented rationale if direct runs are infeasible.

**Questions:**

- Can you report some features of the different models, such as parameters, FLOPs, and memory for your method vs CFSE/CaFo/BP for better comparison?

**Details Of Ethics Concerns:**

This has been flagged in order for the Program Chairs/Area Chair to look into this as I believe their might be indications for plagiarism with prior work mentioned below.

Misrepresentation of novelty and incomplete attribution of prior work points to poor research integrity practices
Manuscript: “Forward-Forward Learning with [SN sets]” (ICLR)
Prior work: Papachristodoulou et al., AAAI 2024 (first introduces CFSE / CwC, PvN, CwC_CE),

In general while the work of AAAI is references in this ICLR submission many components are claimed as new, however they where originally introduced in the prior work.

Identical loss functions misrepresented as distinct:
The ICLR submission’s “CwC” and “CE” losses are mathematically and implementation-wise identical-both apply softmax cross-entropy to the same class-wise logits. The paper, however, treats them as different objectives in text and results, thereby claiming the same novelty.

Unattributed reuse of the channel-wise PvN adaptation:
The submission’s PvN formulation, code structure, and descriptive text closely replicate the AAAI 2024 implementation and exposition (channel-wise positive/negative margins with threshold), but credit only Hinton’s original Forward-Forward algorithm, omitting the AAAI adaptation source.

Documented code and textual parallels:
Side-by-side code excerpts from the AAAI and ICLR repositories show line-level equivalence across all three losses (CwC manual, CE, PvN), differing only in variable names thus potentially demonstrating that the same codebase was used but the implementation for the original losses where not acknowledged correctly.

Code in supplementary material has been examined and detailed point by point analysis can also be provided.

---

> ### Author Response · Authors · 2025-11-22
>
> Thank you very much for your comments. Before providing our point-to-point response, we would like to clarify that we have already submitted a detailed reply regarding the concerns about plagiarism directly to the PC and AC. Below we try to address your concerns.
>
> **$\textbf{Comment: }$**
>
> Conv-layer partitioning into sets is a mechanism already introduced in prior work CFSE (Papachristodulou et al, 2024). The paper acknowledges this but still lists this set partitioning as a core contribution. In my opinion the genuine novelty is the linear-layer extension and width adaptation. In addition, all three losses have already been introduced in prior work with the same name (Papachristodulou et al, 2024). The PvN and CwC as used here are already defined in CFSE (Papachristodulou et al, 2024) for channel groups (CFSE Eqs. 5–6). The manuscript should be explicit about reuse and position its difference. As it stands the mathematical formulation is the same.
>
> **$\textbf{Response:}$**
>
> There may be a misunderstanding caused by insufficient clarity in our manuscript. Our intention was to convey that the proposed concept of SN sets is inspired by and generalizes the grouping idea introduced in the AAAI paper. This concept further enables the design of the growth and degeneration mechanisms. The definition of “neurons” can vary depending on the model architecture: they may refer to individual elements within a latent feature vector (as in linear layers), channels within feature maps (as in convolutional layers), or even spatial units such as pixels within feature maps.
>
> Regarding convolutional layers, we have also tested several partitioning strategies, including flattening feature maps into vectors and then splitting SN sets in a manner identical to the linear layer. However, the channel-wise partition performed the best. We apologize for the lack of clarity and will revise the manuscript thoroughly to avoid further confusion.
>
> **$\textbf{Comment:}$**
>
> The paper’s CwC equals CE applied to per-class goodness logits (g_l). The CE section repeats the same. This duplication confuses the contribution and makes the loss study harder to interpret.
>
> **$\textbf{Response:}$**
>
> It is true that CWC becomes identical to CE when paired with softmax. However, in practice, CWC involves several stability-oriented tuning steps, as also described in the AAAI work. These practical differences can lead to the performance discrepancies observed in our experiments.
>
> Additionally, one may adopt alternatives to softmax, and we plan to provide further experiments using such alternatives and revise manuscript to avoid any ambiguity.
>
> **$\textbf{Comment:}$**
>
> Section 3.7 claims training each layer “directly on the entire dataset” and that optimization is “essentially closed-form.” The appendix algorithm trains with random mini-batches using SGD/Adam, i.e., not closed-form nor full-batch.
>
> **$\textbf{Response: }$**
>
> Our training procedure differs substantially from that of the AAAI work, as shown in Section 3.7:
> ```
> Ours:
> For layers:
>     For epoch:
>         For minibatch:
>             Train layer i
> Theirs:
> For epoch:
>     For minibatch:
>         For layers:
>             Train layer i
> ```
> This is why we state that each layer is trained using the entire dataset: we apply SGD or Adam to optimize a given layer across the full dataset, after which we proceed to train the next layer.
>
> By “closed-form,” we mean that the optimization problem becomes analytically tractable once the loss function is fixed, because at each stage only a single layer requires optimization. This structure makes it considerably easier to design and analyze the training procedure compared to jointly training all layers of a neural network.
>
> **$\textbf{Comment:}$**
>
> Only last-layer edits, small additions/deletions, and sensitivity (including degradations) are shown. Authors note single runs with fluctuating results in some cases.
>
> **$\textbf{Response:}$**
>
> As discussed in Section 5, the latent growth and degeneration mechanisms require further work to be fully developed. Nonetheless, we hope that our preliminary investigation can offer useful insights for future research in this direction. Due to space limitations, we reported only single-run results in the main text; additional runs are provided in the Appendix.
>
> **$\textbf{Comment: }$**
>
> Reported gains over CFSE/CaFo are rather incremental (e.g., CIFAR-10 80.5% vs CFSE 77.2%; CIFAR-100 52.0% vs CFSE 48.9%). This is fine, but it does not show a qualitative jump.
>
> **$\textbf{Response:}$**
>
> We agree that there is no qualitative jump. Achieving such a jump is indeed difficult. However, beyond the performance improvements, our layer-by-layer training framework provides flexibility in depth: training may stop early if the current layer already achieves satisfactory performance, or additional layers may be added after the current model is trained.

---

> ### Author Response · Authors · 2025-11-22
>
> **$\textbf{Comment: }$**
>
> The paper states “architectures are kept consistent” in Table 2 comparisons, but also evaluates CaFo with a revised architecture (Re). This change should be labeled carefully.
>
> **$\textbf{Response: }$**
>
> We will make this point explicit in the revised manuscript.
>
> **$\textbf{Comment: }$**
>
> The paper lacks comparison and/or discussion with standard local-learning and FF-inspired methods beyond CFSE/CaFo. In particular FA, DFA, DRTP, SoftHebb. These are canonical non-BP or local-error baselines that target similar motivations (forward compute, cheap feedback, biological plausibility). The paper is missing a comparison with at least one method per family or provide a documented rationale if direct runs are infeasible.
>
> **$\textbf{Response: }$**
>
> The comparison experiments and several other analyses are currently in progress, and we will submit all results together with the revised manuscript as soon as they are completed.
> Moreover, we would like to clarify that we will carefully revise the manuscript— including rephrasing the description of our contributions—and further strengthen the acknowledgment of related work, especially the referenced AAAI paper, to avoid any possible misunderstanding or confusion regarding plagiarism.
>
> Regarding the code, particularly the implementation of the loss functions, although we have already provided supporting evidence to the AC and PC, we would like to emphasize that we have never intended to present these losses as our own contributions. The implementation is not claimed as a novelty. Since these loss functions are standard and their code sources are open in the communicty, it is quite common for different implementations to appear similar when they serve the same mathematical purpose.

---

> ### Author Response · Authors · 2025-11-25
>
> According to your comments, we have conducted additional experiments and revised the manuscript accordingly. Specifically, we clarified the contributions to avoid potential confusion and strengthened the discussion of related work, particularly the AAAI paper. We also clarified the differences between CE loss and CWC loss and added an additional experiment on CE with the Sparsemax function, an alternative to Softmax. Furthermore, we further emphasized the distinctions between our work and the AAAI paper in Section 3.7. Several minor revisions, including updates to the experimental settings, have also been incorporated. Also, we have done several works according to other reviewers’ comments.
> All revisions are highlighted in blue for your convenience, and the color will be removed in the final submission. The main revisions are summarized as follows:
> 1. We revised parts of the main manuscript to give proper credit to the references and to further clarify our contributions, aiming to avoid potential confusion. .   Besides, we slightly revised the training pseudo code for clarity.
> 2. We added additional experiments to visualize latent activations across all layers (Appendix D.1). These include class-wise response analysis on MNIST and CIFAR-10, as well as cross-class behavior analysis among several highly related classes on CIFAR-100. We believe these results enhance the explainability of neuron behaviors.
> 3. We added further experiments on neuron growth and degeneration (Appendix D.2). First, we visualized neuron responses after growth to analyze the behavior of new neurons. Second, we observed that deep layers contain more inactive neurons than the first layer, particularly in the CIFAR-10 experiments. We further illustrate overall accuracy versus degenerated neurons to demonstrate the practical impact of degeneration.
> 4. We added experiments examining the training dynamics, including loss values and test accuracy (Appendix D.3). These results not only depict the training behavior but also support the previous conclusion that the PvN loss is more suitable for linear layers, while the CWC loss is more suitable for convolutional layers. Due to computational constraints, these results are based on a single run; we plan to replace them with averages over multiple runs in a future version.
> 5. We added experiments for a new loss, CE with Sparsemax function, to further distinguish it from the CWC loss (Section 4.2). In addition, we corrected a mis-recording of the CaFo method and revised the MNIST architecture accordingly.
> 6. We added experiments on convolutional version of FF (in Section 4.3).
>
> The above results are not exhaustive. Ongoing work includes:
>
> 1. Accuracy evaluation of other comparison algorithms, including non-BP algorithms and FF variants.
>
> 2. Sensitivity analysis for certain hyperparameters.
>
> Furthermore, we plan to include additional details in the manuscript, such as pseudocode for the growth and degeneration procedures.
>
> We submit the current revised version, which does not yet include all results, with the goal of demonstrating that the newly added results significantly improve the quality of the paper. As there is still time before the rebuttal deadline, we hope this allows reviewers sufficient time for evaluation. We apologize for the delay in the revised manuscript and regret any potential inconvenience this may cause to the reviewers.

---

> ### Author Response · Authors · 2025-11-28
>
> We have updated the fully revised manuscript according to the reviewers’ valuable comments. Specifically, compared to the previous version, we have added new comparison results for FF variants (e.g., DeeperForward) as well as non-BP and biologically implausible methods (DRTP and BWBPF). Moreover, we have included sensitivity analyses on both the PvN threshold and the degeneration threshold. Overall, the sensitivity analysis shows that although the PvN threshold influences performance, the resulting fluctuations remain small and layer-dependent. In addition, the results on the degeneration threshold indicate that overparameterized networks contain many weak or redundant neurons, underscoring the potential of our degeneration-based approach.

---

### Official Review · Reviewer_9LLh · 2025-11-01

**Soundness:** 4
**Presentation:** 3
**Contribution:** 3
**Rating:** 6
**Confidence:** 3

**Summary:**

The paper introduces an enhancement to the standard Forward-Forward (FF) algorithm designed specifically for classification tasks. It addresses a key limitation of the original FF model and its reliance on high-quality negative samples. They propose the concept of Support Neuron (SN) sets and integrating Dynamic Architecture Adaptation using grow and prune neurons concept. The SN sets partition neurons in each layer to explicitly correspond to class concepts, thus enforcing local class-specific representations and improving feature discrimination compared to generic FF variants.

**Strengths:**

- Directly solves a major weakness of the original FF algorithm, making the training process more robust and easier to implement without complex negative sample generation strategies.

- The explicit partitioning of neurons into class-specific SN sets creates a more structured and potentially more interpretable feature space, as one can directly analyze the activity of neurons associated with a particular class.
- The paper is well written and easy to follow.

**Weaknesses:**

- Computational overhead of the Dynamic Architecture Adaptation is unknown compared to the standard FF models.
Provide a detailed analysis of the average time increase per training epoch (or step) compared to a standard, fixed-architecture FF model with the same number of layers and total neurons.


- The core concept relies on partitioning neurons into explicit class-based sets. This is straightforward for simple classification (e.g., CIFAR, MNIST) but breaks down for tasks without fixed, discrete classes limiting the approach's general applicability. How easy or difficult to extend this beyond classification is not known.

- The rigid, class-specific partitioning of neurons into SN sets, while enhancing interpretability, might constrain the model's ability to learn efficient, shared representations. For fine-grained classification, this forced disentanglement could be less efficient than a flexible, dense layer that naturally compresses shared features across multiple classes. Provide emprical results for fine grained classification and use Centered Kernel Alignment (CKA) to quantify the similarity between the feature representations of two highly related classes (e.g., two different bird species or types of cars) in the penultimate layer. Compare the CKA score between the Dynamic FF model and a standard, dense FF model.

- While the Support Neuron (SN) sets are designed to mitigate the reliance on high-quality negative samples, the paper might not eliminate the need for them entirely.  Compare the Dynamic SN-FF model's performance against the standard FF baseline when both are trained using trivial, low-quality negative samples (e.g. completely random noise or samples drawn from a different distribution).

- Usually, dynamically adjusting the architecture introduces instability during training. Any insights or detailed discussion regarding the model training will be useful.

**Questions:**

See Weakness section

---

> ### Author Response · Authors · 2025-11-22
>
> Thank you very much for your valuable comments. Below we provide our point-to-point response.
>
> **$\textbf{Comment:}$**
>
> “Computational overhead of the Dynamic Architecture Adaptation is unknown compared to the standard FF models. Provide a detailed analysis of the average time increase per training epoch (or step) compared to a standard, fixed-architecture FF model with the same number of layers and total neurons.”
>
> **$\textbf{Response:}$**
>
> There may be a misunderstanding. Our proposed method operates on a well-trained layer. We first train a fixed-architecture layer; if its performance is unsatisfactory, we then activate the growth mechanism to introduce additional neurons. The training of these new neurons is isolated and does not require retraining the already-trained parts of the layer. Therefore, the computational overhead of dynamic architecture adaptation depends only on the growth and degeneration procedures.
>
> Under the settings described in Section 4.4, the training times on MNIST are 2.146 s for growth and 0.778 s for degeneration. On CIFAR-10, the corresponding times are 3.499 s and 3.674 s.
>
> **$\textbf{Comments:}$**
>
> The core concept relies on partitioning neurons into explicit class-based sets. This is straightforward for simple classification (e.g., CIFAR, MNIST) but breaks down for tasks without fixed, discrete classes limiting the approach's general applicability. How easy or difficult to extend this beyond classification is not known.
>
> **$\textbf{Response:}$**
>
> It is true that our initial training strategy supports only a fixed-class setting. However, the neuron-growth mechanism can be applied to generate new neurons for classes not present in the initial training set. That said, extending the method to this setting requires further investigation, since both the newly added neurons and the existing neurons must learn information from the newly introduced class data. We have tested the original growth method but failed. This direction may warrant further investigation.
>
> **$\textbf{Comment: }$**
>
> The rigid, class-specific partitioning of neurons into SN sets, while enhancing interpretability, might constrain the model's ability to learn efficient, shared representations. For fine-grained classification, this forced disentanglement could be less efficient than a flexible, dense layer that naturally compresses shared features across multiple classes. Provide empirical results for fine grained classification and use Centered Kernel Alignment (CKA) to quantify the similarity between the feature representations of two highly related classes (e.g., two different bird species or types of cars) in the penultimate layer. Compare the CKA score between the Dynamic FF model and a standard, dense FF model.
>
> **$\textbf{Response:}$**
>
> In our method, SN sets are used solely to determine neuron-to-class assignments. Practically, they are involved in loss computation and prediction. The inter-layer connections are fully dense: a neuron assigned to SN set i is still connected to neurons assigned to SN set j (for all i ≠ j) in the previous layer.
>
> Allowing overlapping neurons across SN sets is indeed reasonable. However, manually designing such overlaps is infeasible. First, every pair of SN sets requires an overlap, and the design becomes intractable when the number of classes is large. Second, it is inappropriate to enforce equal overlap sizes for all class pairs — for example, the overlap between “white cat” and “black cat” should naturally be larger than that between “white cat” and “white car.”
>
> For these reasons, an automatic method is needed. We attempted the following reorganization algorithm:
> ```
> Reorganization (per layer)
> For each class k among the K classes:
> • Input class-k data to the layer
> • Record each neuron’s average response
> • Record the average response of the k-th SN set
> • If a neuron not in SN set k shows an average response higher than the SN-set average, add it to SN set k
> ```
>
> Unfortunately, this algorithm did not introduce any improvement; in fact, it had a negative effect, whether applied after each layer’s training or integrated into the training process. There may exist more effective approaches to constructing overlaps, and we highlight this as a promising future direction in Section 5 (Conclusion and Perspectives).
>
> Regarding the CKA experiments, we are currently conducting a series of evaluations to visualize intermediate activations or quantify representation similarity. We will submit these results along with the revised manuscript as soon as they are finalized.

---

> ### Author Response · Authors · 2025-11-22
>
> **$\textbf{Comment: }$**
>
> While the Support Neuron (SN) sets are designed to mitigate the reliance on high-quality negative samples, the paper might not eliminate the need for them entirely. Compare the Dynamic SN-FF model's performance against the standard FF baseline when both are trained using trivial, low-quality negative samples (e.g. completely random noise or samples drawn from a different distribution).
>
> **$\textbf{Response: }$**
>
> There may be a misunderstanding. In the FF algorithm, positive and negative samples are required to compute goodness values for the loss and prediction. In contrast, our work follows the grouping idea of Papachristodulou et al. (2024): the positive and negative goodness values are replaced by the responses of the corresponding SN sets. This eliminates the need for explicit positive/negative samples. In other words, we compute goodness using SN-set responses rather than sample-level positive/negative data.
>
> **$\textbf{Comments:}$**
>
> Usually, dynamically adjusting the architecture introduces instability during training. Any insights or detailed discussion regarding the model training will be useful.
>
> **$\textbf{Response: }$**
>
> As noted in point 1, training new neurons does not require retraining the whole layer. Thus, the stability of the growth process depends solely on the design of the growth algorithm (degeneration requires no training). We provided additional experiments in Appendix D to illustrate the impact and stability of growth on accuracy.

---

> ### Author Response · Authors · 2025-11-25
>
> According to your valuable comments, we have conducted additional experiments and revised the manuscript accordingly. In particular, we now provide visualizations of neuron behaviors across all layers for several closely related classes (i.e., bottle, bowl, can, and cup), as shown in Appendix D.1. Also, we have done several works according to other reviewers’ comments.
> All revisions are highlighted in blue for your convenience, and the color will be removed in the final submission. The main revisions are summarized as follows:
> 1. We revised parts of the main manuscript to give proper credit to the references and to further clarify our contributions, aiming to avoid potential confusion.   Besides, we slightly revised the training pseudo code for clarity.
> 2. We added additional experiments to visualize latent activations across all layers (Appendix D.1). These include class-wise response analysis on MNIST and CIFAR-10, as well as cross-class behavior analysis among several highly related classes on CIFAR-100. We believe these results enhance the explainability of neuron behaviors.
> 3. We added further experiments on neuron growth and degeneration (Appendix D.2). First, we visualized neuron responses after growth to analyze the behavior of new neurons. Second, we observed that deep layers contain more inactive neurons than the first layer, particularly in the CIFAR-10 experiments. We further illustrate overall accuracy versus degenerated neurons to demonstrate the practical impact of degeneration.
> 4. We added experiments examining the training dynamics, including loss values and test accuracy (Appendix D.3). These results not only depict the training behavior but also support the previous conclusion that the PvN loss is more suitable for linear layers, while the CWC loss is more suitable for convolutional layers. Due to computational constraints, these results are based on a single run; we plan to replace them with averages over multiple runs in a future version.
> 5. We added experiments for a new loss, CE with Sparsemax function, to further distinguish it from the CWC loss (Section 4.2). In addition, we corrected a mis-recording of the CaFo method and revised the MNIST architecture accordingly.
> 6. We added experiments on convolutional version of FF (in Section 4.3).
>
> The above results are not exhaustive. Ongoing work includes:
>
> 1. Accuracy evaluation of other comparison algorithms, including non-BP algorithms and FF variants.
>
> 2. Sensitivity analysis for certain hyperparameters.
> Furthermore, we plan to include additional details in the manuscript, such as pseudocode for the growth and degeneration procedures.
>
>
> We submit the current revised version, which does not yet include all results, with the goal of demonstrating that the newly added results significantly improve the quality of the paper. As there is still time before the rebuttal deadline, we hope this allows reviewers sufficient time for evaluation. We apologize for the delay in the revised manuscript and regret any potential inconvenience this may cause to the reviewers.

---

> ### Author Response · Authors · 2025-11-28
>
> We have updated the fully revised manuscript according to the reviewers’ valuable comments. Specifically, compared to the previous version, we have added new comparison results for FF variants (e.g., DeeperForward) as well as non-BP and biologically implausible methods (DRTP and BWBPF). Moreover, we have included sensitivity analyses on both the PvN threshold and the degeneration threshold. Overall, the sensitivity analysis shows that although the PvN threshold influences performance, the resulting fluctuations remain small and layer-dependent. In addition, the results on the degeneration threshold indicate that overparameterized networks contain many weak or redundant neurons, underscoring the potential of our degeneration-based approach.

---

### Author Response · Authors · 2025-12-03
**Rebuttal Summary for the Area Chair**

We acknowledge the updated rebuttal rules and understand that, due to system issues, reviewers are no longer able to respond. For the convenience of the new Area Chair, we provide below a brief summary of the reviewers’ primary comments and our corresponding responses.

**$\textbf{Reviewer 1}$**

C1: Concerns about computational overhead and increased training time per epoch caused by dynamic architecture adaptation.

R1: We clarified that adaptation occurs only on already well-trained layers. The computational overhead depends solely on the Growth and Degeneration procedures. Additionally, we reported training times under default settings: Growth—2.146 s on MNIST and 3.499 s on CIFAR-10; Degeneration—0.778 s on MNIST and 3.674 s on CIFAR-10.

C2: The core concept relies on explicit class-based partitioning, limiting applicability to tasks without fixed, discrete classes.

R2: We agreed that our current design initially supports fixed-class scenarios. While neuron growth may potentially handle new classes, our preliminary attempts were unsuccessful, indicating that this problem requires further investigation.

C3: Class-specific partitioning might limit the model’s ability to learn shared representations, particularly in fine-grained classification. Requested empirical results to quantify similarity between feature representations.

R3: We explained that inter-layer connections remain fully dense, and SN sets only determine neuron-to-class assignments for prediction. We tested an automatic reorganization algorithm to introduce overlaps, but it negatively impacted performance. We also visualized latent-layer neurons to illustrate their behavior on groups of closely related classes.

C4: The method may not fully eliminate the need for high-quality negative samples. Requested  a comparison against the standard FF baseline using trivial, low-quality negatives.

R4: We clarified that our method does not rely on explicit positive/negative samples. Goodness evaluation is based entirely on SN-set responses.

C5: Dynamically adjusting the architecture may introduce instability during training. Requested discussion on stability.

R5: We clarified that stability depends on the design of the growth algorithm, as new neurons are trained locally without retraining entire layers. Additional experiments in Appendix D demonstrate the stability and effects of growth and degeneration.

**$\textbf{Reviewer 2}$**

C1: Convolutional-layer partitioning already exists in CFSE (AAAI). The manuscript lists it as a core contribution, but genuine novelty lies in linear-layer extension and width adaptation. All three losses (PvN, CwC) are defined in CFSE. Manuscript should clearly distinguish reuse from novelty.

R1: We clarified that SN sets generalize the grouping idea from the AAAI paper while enabling growth and degeneration. We further clarified potential misunderstandings regarding the term “neurons” (elements, channels, pixels) and tested alternative partitioning strategies. The manuscript was revised to eliminate ambiguity.

C2: CwC is equivalent to CE applied to per-class goodness logits, leading to redundancy and potential confusion.

R2: We agreed that CwC equals CE when paired with softmax. However, in practice, CwC includes stability-oriented adjustments that result in observable performance differences. We also added experiments applying CE with Sparsemax for additional clarification.

C3: Section 3.7 claims training each layer “directly on the entire dataset” and optimization is “essentially closed-form,” but the code shows random mini-batches (SGD/Adam).

R3: We clarified that “layer-by-layer training across the full dataset” refers to sequentially training one layer at a time, and “closed-form” refers to the analytical tractability of single-layer optimization.

C4: Only last-layer edits, small structural changes, and single-run results are shown.

R4: We explained that latent-layer growth/degeneration requires further investigation. Additional runs and extended experiments are provided in the Appendix.

C5: Reported gains over CFSE/CaFo are incremental (e.g., CIFAR-10 80.5% vs 77.2%) and do not represent a qualitative leap.

R5: We agreed there is no qualitative jump and emphasized that the layer-by-layer training framework provides both improved flexibility and competitive accuracy.

C6: Paper states “architectures are kept consistent,” but CaFo was evaluated with a revised architecture. Requested clearer labeling.

R6: We revised the corresponding section to ensure clarity.

C7: Lacks comparison/discussion with standard local-learning and FF-inspired methods beyond CFSE/CaFo (FA, DFA, DRTP, SoftHebb).

R7: We added comparisons with FF variants (DeeperForward) and non-backpropagation methods (DRTP, BWBPF).

We also provided evidence addressing the plagiarism concern and raised a concern regarding potential bias (visible only to AC and PC).

---

### Author Response · Authors · 2025-12-03
**Rebuttal Summary for the Area Chair**

**$\textbf{Reviewer 3}$**

C1: Motivation for neuron growth and degeneration lacks rigorous justification or toy experiment validation.

R1: We explained that the motivation is biologically inspired, aiming to inspire further exploration of brain-inspired modeling.

C2: DNN baseline on CIFAR-100 (65.4%) seems low; requested full architecture and training details.

R2: We provided the complete architecture and optimization setup, emphasizing that the baseline was intentionally selected to enable fair comparison under identical training conditions.

C3: Requested thorough analysis of neuron removal: (a) per-neuron activation statistics, (b) visualization of filters/activations, (c) results averaged across multiple seeds.

R3: We clarified that degeneration is sequential, and averaging class-wise results across runs is inappropriate due to structural differences. We added new visualizations of neuron responses across layers (Appendix D.1).

C4: Asked whether alternative SN-set response metrics (e.g., L1 norm, max pooling, cosine similarity) were tested.

R4: We reported tests with L1 norm, SN-set-wise max pooling, and random SN-set selection.

C5: Requested stronger BP baselines (well-tuned DNN/ResNet) and recent FF variants (DeeperForward/Distance-Forward).

R5: We clarified that the main goal is demonstrating the proposed method under identical settings. Additional experiments on other FF variants and non-BP methods were included.

C6: Requested sensitivity study for key hyperparameters: PvN threshold, growth selection rules, neurons added per step, degeneration thresholds.

R6: We included sensitivity analyses for PvN and degeneration thresholds, clarifying that new neurons rely heavily on existing ones, so increasing neurons per growth step does not improve performance.

**$\textbf{Reviewer 4}$**

C1: Inconsistencies between reported CaFo results and original paper.

R1: We clarified that differences arise from architectural and training protocol adjustments.

C1 (follow-up): Raised concern about fairness due to not reporting original CaFo MNIST architecture.

R1 (follow-up): We explained that the goal was to evaluate linear layers (only on MNIST) and convolutional layers (only on CIFAR). The original convolutional CaFo MNIST architecture was modified to a fully connected network for fair comparison.

C2: Lack of visual/statistical evidence of training dynamics and representation evolution (accuracy plots, activation heatmaps, SN-set response distributions).

R2: We added visualizations of training dynamics, neuron activations across layers, and their evolution after growth/degeneration.

C3: Explicit static, class-wise SN assignment lacks theoretical justification; may harm generalization.

R3: We agreed that overlapping SN sets are theoretically reasonable, but manual design is infeasible. Automatic Reorganization was attempted but degraded performance.

C3 (follow-up): Recommended further analysis of neuron Reorganization.

R3 (follow-up): We investigated why Reorganization failed: reorganized neurons only responded correctly to a small portion of new-class data, decreasing accuracy.

C4: Unclear if Combo training (CwC for conv + PvN for linear) is sequential or simultaneous.

R4: We clarified that training is sequential, layer-by-layer, allowing different loss functions per layer. Pseudocode is provided in Algorithm 1.

C5: FF only tested on MNIST.

R5: We added convolutional FF experiments on CIFAR, along with comparisons against FF variants and non-BP methods.

---

### Author Response · Authors · 2025-12-03
**Summary of Major Revisions**

We sincerely appreciate the effort that all reviewers have invested, which has significantly improved the quality of our paper. The final manuscript is submitted and below we provide a summary of the major revisions.

1.	We revised parts of the main manuscript to give proper credit to the references and to further clarify our contributions, aiming to avoid potential confusion. Besides, we slightly revised the training pseudo code for clarity.

2.	We added additional experiments to visualize latent activations across all layers (Appendix D.1). These include class-wise response analysis on MNIST and CIFAR-10, as well as cross-class behavior analysis among several highly related classes on CIFAR-100. We believe these results enhance the explainability of neuron behaviors.

3.	We added further experiments on neuron growth and degeneration (Appendix D.2). First, we visualized neuron responses after growth to analyze the behavior of new neurons. Second, we observed that deep layers contain more inactive neurons than the first layer, particularly in the CIFAR-10 experiments. We further illustrate overall accuracy versus degenerated neurons to demonstrate the practical impact of degeneration.

4.	We added experiments examining the training dynamics, including loss values and test accuracy (Appendix D.3). These results not only depict the training behavior but also support the previous conclusion that the PvN loss is more suitable for linear layers, while the CWC loss is more suitable for convolutional layers. Due to computational constraints, these results are based on a single run; we plan to replace them with averages over multiple runs in a future version.

5.	We added experiments for a new loss, CE with Sparsemax function, to further distinguish it from the CWC loss (Section 4.2). In addition, we corrected a mis-recording of the CaFo method and revised the MNIST architecture accordingly.

6.	We added experiments on the convolutional version of FF and new comparison results for FF variants (e.g., DeeperForward) as well as non-BP and biologically implausible methods (DRTP and BWBPF) (Section 4.3). Moreover, we revised this section for improved clarity.

7.	We have included sensitivity analyses on both the PvN threshold and the degeneration threshold. Overall, the sensitivity analysis shows that although the PvN threshold influences performance, the resulting fluctuations remain small and layer-dependent. In addition, the results on the degeneration threshold indicate that overparameterized networks contain many weak or redundant neurons, underscoring the potential of our degeneration-based approach.

---

### Meta-Review · Area_Chair_nMaK · 2025-12-29

**Summary:**

This submission proposes a Forward-Forward variant for classification that partitions each layer into class-specific Support Neuron sets, defines class “goodness” via SN-set responses, and adds dynamic width adaptation through neuron growth and degeneration, with experiments on MNIST/CIFAR and comparisons to CFSE/CaFo and limited BP baselines. Reviewers’ concerns center on whether the technical novelty is sufficiently distinct from CFSE—especially for convolutional-layer grouping and the reuse of named losses—along with confusing presentation around loss definitions, questionable claims about “closed-form”/full-dataset training versus the actual minibatch optimization, and weaknesses in baseline fairness and coverage. Multiple reviewers also highlighted that the growth/degeneration mechanisms and static class-wise assignment are only weakly justified, and that the empirical story relies heavily on limited analyses and incremental gains that do not yet establish a compelling qualitative advance. While one reviewer is mildly positive about the practical framing  and interpretability angle, the overall set of concerns—particularly novelty attribution, methodological clarity, and evaluation completeness—collectively support a weak reject recommendation.

**Reviewer Concerns:**

The rebuttal and revisions partially improve clarity and address some concrete requests, but several key issues remain. On the positive side, the authors clarified that training is sequential layer-by-layer (resolving the “Combo” ambiguity), added training-dynamics plots and neuron-activation visualizations, expanded comparisons to additional FF variants and some non-BP baselines, and provided sensitivity analyses for selected thresholds; they also acknowledged and corrected reporting issues (e.g., CaFo training-epoch mismatch) and revised text to better attribute prior work and remove ambiguous wording. However, the core novelty positioning relative to CFSE remains insufficiently convincing given that the conv-layer partitioning and named losses substantially overlap with prior work and the manuscript’s contribution framing still appears easy to misread as broader novelty than supported, and the CwC-vs-CE discussion remains conceptually muddy despite attempts to justify “practical differences.” More importantly, concerns about baseline fairness and completeness are not fully resolved: the justification for modifying CaFo on MNIST (switching to FC to emphasize linear layers) does not fully eliminate the perception of selective comparison design, and the broader baseline set is still not at the level expected for a decisive ICLR paper in this area. Finally, the “biologically inspired” motivation for growth/degeneration and the static class-wise SN assignment are still largely heuristic, and the added analyses (often limited runs or constrained settings) do not yet establish robustness, generality beyond fixed-class classification, or a clear mechanism-level understanding of when and why growth/degeneration helps; as a result, the rebuttal improves presentation but does not fully clear the main technical and empirical bars.

**Reviewer Scores:**

Reviewer 5spw would likely remain at a strong reject given their high-confidence view that the novelty attribution and loss framing are still too close to CFSE and that the evaluation and claims remain misaligned with what is genuinely new, even after added citations and clarifications. Reviewer GDeD might move slightly upward from reject to a clearer but still negative stance, as the sequential training ambiguity is resolved and additional comparisons/visualizations are added, yet their concerns about selective baseline reporting and the theoretical rationale for explicit class-wise assignment remain only partially addressed. Reviewer Hfab could shift marginally toward the middle (e.g., from marginally below threshold to a firmer borderline), since several of their concrete requests (training details, alternative metrics, added visualizations, partial sensitivity and added baselines) were addressed, but they would still likely keep the paper below the bar due to ad hoc motivation and limited robustness/variance evidence. Reviewer 9LLh, already mildly positive, might stay near their original borderline-above-threshold position or soften slightly given the unresolved generality and baseline-strength concerns, but would not be enough to offset the remaining high-confidence negative assessment and the still-incomplete substantiation of novelty and evaluation, supporting an overall weak reject.

---

### Decision · Program_Chairs · 2026-01-26

Reject